# FAST SUMMATION OF RADIAL KERNELS VIA QMC SLICING

**Johannes Hertrich**[1]**, Tim Jahn**[2]**, Michael Quellmalz**[2]

[1] Université Paris Dauphine-PSL and UCL, `johannes.hertrich@dauphine.psl.eu`
[2] Technische Universität Berlin, {`jahn, quellmalz`}`@math.tu-berlin.de`

## ABSTRACT

The fast computation of large kernel sums is a challenging task, which arises as a subproblem in any kernel method. We approach the problem by slicing, which relies on random projections to one-dimensional subspaces and fast Fourier summation. We prove bounds for the slicing error and propose a quasi-Monte Carlo (QMC) approach for selecting the projections based on spherical quadrature rules. Numerical examples demonstrate that our QMC-slicing approach significantly outperforms existing methods like (QMC-)random Fourier features, orthogonal Fourier features or non-QMC slicing on standard test datasets.

## 1 INTRODUCTION

We consider fast algorithms for computing the kernel sums

$$s_m = \sum_{n=1}^{N} w_n K(x_n, y_m), \quad \text{for all} \quad m = 1, ..., M, \tag{1}$$

where $x_n, y_m \in \mathbb{R}^d$ and $w_n \in \mathbb{R}$ for $n = 1, ..., N$, $m = 1, ..., M$ and $K \colon \mathbb{R}^d \times \mathbb{R}^d \to \mathbb{R}$ is a radial kernel, i.e., $K(x, y) = F(\|x - y\|)$ for the Euclidean norm $\|\cdot\|$ and some $F \colon \mathbb{R}_{\geq 0} \to \mathbb{R}$. This summation problem appears in most kernel methods including kernel density estimation (Parzen, 1962; Rosenblatt, 1956), classification via support vector machines (Steinwart & Christmann, 2008), dimensionality reduction with kernelized principal component analysis (Schölkopf & Smola, 2002; Shawe-Taylor & Cristianini, 2004), distance measures on the space of probability measures like maximum mean discrepancies or the energy distance (Gretton et al., 2006; Székely, 2002), corresponding gradient flows (Arbel et al., 2019; Galashov et al., 2024; Hagemann et al., 2024; Hertrich et al., 2024; Kolouri et al., 2022), and methods for Bayesian inference like Stein variational gradient descent (Liu & Wang, 2016). Computing (1) exactly for all $m = 1, ..., M$ has complexity $\mathcal{O}(MN)$, which can be restricting if $M$ and $N$ are large.

In low dimensions, there is a rich literature on fast approximation algorithms, we include a (non-exhaustive) list in the "related work" section. One particular approach is the fast Fourier summation (Kunis et al., 2006; Potts et al., 2004), which approximates the kernel by a truncated Fourier series and reformulates (1) using the fast Fourier transform on non-equispaced data (Beylkin, 1995; Dutt & Rokhlin, 1993). We provide a short overview in Appendix I. This kind of methods usually provides a computational complexity of $\mathcal{O}(M + N + N_{\mathrm{ft}} \log(N_{\mathrm{ft}}))$, where $N_{\mathrm{ft}}$ is the number of relevant Fourier coefficients, and admits very fast error rates for $N_{\mathrm{ft}} \to \infty$ (even exponential if the kernel is sufficiently smooth). However, the number $N_{\mathrm{ft}}$ of relevant Fourier coefficients grows exponentially with the dimension $d$, such that this method is computationally intractable for dimensions larger than four.

As a remedy for higher dimensions, Rahimi & Recht (2007) proposed random Fourier features (RFF). They represent a positive definite kernel via Bochner's theorem (Bochner, 1933) as the Fourier transform of a non-negative measure. Sampling randomly from this measure at $D \in \mathbb{N}$ points (features) leads to an approximation algorithm with computational complexity $\mathcal{O}(D(N+M))$ independent of the dimension $d$. However, the error decays only with rate $\mathcal{O}(1/\sqrt{D})$, which can be limiting if a high accuracy is required. Moreover, RFF are limited to positive definite kernels and

do not apply to other kernels, like the negative distance kernel $K(x, y) = -\|x - y\|$, which has applications, e.g., within the energy distance (Székely, 2002) that is used for defining a distance on the space of probability measures.

A related approach is slicing (Hertrich, 2024), which represents the kernel sum (1) as an expectation of one-dimensional kernel sums of the randomly projected data points with a different kernel. Discretizing the expectation by sampling at $P$ random projections, the kernel sums (1) can be approximated by $P$ one-dimensional kernel sums, which can be computed efficiently, e.g., by fast Fourier summation. Similarly as for RFF, this leads to a complexity of $\mathcal{O}(P(N + M + N_{\mathrm{ft}} \log N_{\mathrm{ft}}))$, where the expected error can be bounded by $\mathcal{O}(1/\sqrt{P})$. For positive definite kernels, a close relation between RFF and slicing was established by Rux et al. (2025), see the short overview in Appendix G. One advantage of slicing is the applicability to kernels that are not positive definite.

A way to improve the $\mathcal{O}(1/\sqrt{P})$ rate is to replace the uniformly chosen directions with specific sequences of points. This yields so-called quasi-Monte Carlo (QMC) algorithms on the sphere, see (Brauchart et al., 2014). Note that there also exist QMC approaches for RFF (Avron et al., 2016). However, they depend on the restrictive assumption that the measure from Bochner's theorem decouples over the dimension, which is true for the Gauss and $L^1$-Laplace[1] kernel, but false for most other common kernels like the Laplace, Matérn or negative distance kernel.

**Contributions**  Our contributions for fast kernel summation in $\mathbb{R}^d$ via slicing are as follows:

- We derive bounds on the slicing error for various kernels in Theorem 1 including all positive definite radial kernels. Furthermore, we exactly compute the variance for the negative distance kernel, the thin plate spline, the Laplace kernel and the Gauss kernel.

- We exploit QMC sequences on the sphere in order to improve the error rate $\mathcal{O}(1/\sqrt{P})$. To ensure the applicability of the QMC error bounds, we prove certain smoothness results for the function which maps a direction $\xi \in \mathbb{S}^{d-1}$ to the corresponding one-dimensional kernel in Theorem 3. The improved error rates are outlined in Corollary 4.

- We conduct extensive numerical experiments on standard test datasets for several kernels and different QMC sequences, and demonstrate that our QMC slicing approach with the proposed distance QMC designs significantly outperforms the non-QMC slicing method as well as (QMC-)RFF. While the advantage of QMC slicing is most significant in dimensions $d \leq 100$, it also performs better in higher dimensions.

**Outline**  In Section 2, we first revisit the slicing approach in detail and present our improved error bounds. Afterwards, in Section 3, we consider quadrature and QMC sequences on the sphere and prove the applicability of the approaches for slicing. We present our numerical results in Section 4 and draw conclusions in Section 5. Additional proofs, plots and evaluations are contained in the appendix. The code for the numerical examples is provided online[2].

RELATED WORK

**Low-Dimensional Kernel Summation**  Fast summation algorithms have been extensively studied in the literature. They include fast kernel summations based on (non-)equispaced fast Fourier transforms (Greengard et al., 2022; Kunis et al., 2006; Potts et al., 2004), fast multipole methods (Beatson & Newsam, 1992; Greengard & Rokhlin, 1987; Lee & Gray, 2008; Yarvin & Rokhlin, 1999), tree-based methods (March et al., 2015a;b) or H- and mosaic-skeleton matrices (Hackbusch, 1999; Minden et al., 2017; Tyrtyshnikov, 1996). For the Gauss kernel, the fast Gauss transform was proposed by Greengard & Strain (1991) and improved by Yang et al. (2003; 2004). More general fast kernel transforms were considered by Ryan et al. (2022).

**QMC and Quadrature on Spheres**  QMC designs on spheres were studied by Brauchart et al. (2014). Here, the quadrature points optimizing the worst-case error in certain Sobolev spaces are given by spherical $t$-designs, which integrate all polynomials up to degree $t$ on the sphere exactly

---

[1]In literature, there exist two versions of the Laplace kernel $K(x, y) = \exp(-\alpha\|x - y\|_1)$ and $K(x, y) = \exp(-\alpha\|x - y\|)$, which differ in the used norm. Since our analysis focuses on radial kernels, we will only consider the second version in the rest of this paper.

[2]available at https://github.com/johertrich/fastsum_QMC_slicing

(Delsarte et al., 1977; Bannai & Bannai, 2009). The construction of spherical designs is highly non-trivial and intractable in high dimensions. For $\mathbb{S}^2$ and $\mathbb{S}^3$, several examples were computed numerically by Gräf & Potts (2011) and Womersley (2018). Gräf et al. (2012) related quadrature rules on the sphere with halftoning problems.

**Sliced Wasserstein Distance**  The idea of slicing is also used in optimal transport. A sliced Wasserstein distance was proposed by Rabin et al. (2012). In contrast to the kernel summation problem, the sliced and non-sliced Wasserstein distance do not coincide and have different properties. QMC rules for the three-dimensional sliced Wasserstein distance were considered by Nguyen et al. (2024).

**Random Fourier Features**  Random Fourier features (RFFs) were proposed by Rahimi & Recht (2007) and were further analyzed in several papers Bach (2017); Hashemi et al. (2023); Li et al. (2021). To improve the error rates, Avron et al. (2016) proposed a quasi-Monte Carlo approach for RFFs under the restrictive assumption that the measure from Bochner's theorem decouples over the dimensions. This approach was refined by Huang et al. (2024) and Munkhoeva et al. (2018). Yu et al. (2016) proposed orthogonal random features. In a very recent preprint, Belhadji et al. (2024) derive an explicit quadrature rule in the Fourier space for efficient summations of the Gauss kernel.

## 2 SLICING OF RADIAL KERNELS

Let $K\colon \mathbb{R}^d \times \mathbb{R}^d$ be a radial kernel of the form $K(x,y) = F(\|x-y\|)$ for some basis function $F\colon \mathbb{R}_{\geq 0} \to \mathbb{R}$. Throughout this paper, we will assume that $K$ has the form

$$K(x,y) = \mathbb{E}_{\xi \sim \mathcal{U}_{\mathbb{S}^{d-1}}}[\mathrm{k}(\langle \xi, x\rangle, \langle \xi, y\rangle)]$$

for some one-dimensional radial kernel $\mathrm{k}\colon \mathbb{R} \times \mathbb{R} \to \mathbb{R}$ with basis function $f\colon \mathbb{R}_{\geq 0} \to \mathbb{R}$, where we suppress the dependence of $f$ and $F$ onto the dimension $d$. By inserting the basis functions of the kernels, this corresponds to the slicing relation

$$F(\|x\|) = \mathbb{E}_{\xi \sim \mathcal{U}_{\mathbb{S}^{d-1}}}[f(|\langle \xi, x\rangle|)]. \tag{2}$$

A pair $(F, f)$ of basis functions in $L_{\mathrm{loc}}^\infty(\mathbb{R}_{\geq 0})$ fulfills this relation if and only if $F$ is the *generalized Riemann–Liouville fractional integral* transform given by

$$F(t) = \frac{2\Gamma(\frac{d}{2})}{\sqrt{\pi}\Gamma(\frac{d-1}{2})} \int_0^1 f(ts)(1-s^2)^{\frac{d-3}{2}}\,\mathrm{d}s, \tag{3}$$

for $2 \leq d \in \mathbb{N}$, see (Hertrich, 2024, Prop 2 and Rubin, 2003). In order to find the one-dimensional basis function $f$ for a given $F$, we have to invert the transform (3). This can be done explicitly if

- i) $F$ is analytic on $\mathbb{R}_{\geq 0}$, i.e., there exists $(a_n)_{n\in\mathbb{N}} \subset \mathbb{R}$ such that $F(x) = \sum_{n=0}^\infty a_n x^n$ for all $x \geq 0$, or
- ii) $F(\|\cdot\|)\colon \mathbb{R}^d \to \mathbb{R}$ is continuous, bounded and positive definite, i.e., for all $N \in \mathbb{N}$, all pairwise distinct $x_j \in \mathbb{R}^d$ and all $a_j \in \mathbb{R}$ for $j = 1, ..., N$ it holds that $\sum_{j,k=1}^N a_j a_k F(\|x_j - x_k\|) \geq 0$,

see (Hertrich, 2024, Thm 3 and Rux et al., 2025, Cor 4.11). We include a list of pairs $(F, f)$ fulfilling (2) in Appendix A. In particular, it includes the basis functions of Gauss, Laplace and Matérn kernels. Moreover, $f$ can be computed for other important choices that fulfill neither i) nor ii), e.g., the thin-plate spline and the generalized Riesz kernel.

### 2.1 FAST KERNEL SUMMATION VIA SLICING

In order to compute the kernel sums (1) efficiently, we approximate $F(\|\cdot\|)$ by projections and the one-dimensional basis function $f$, i.e., we aim to find directions $\xi_1, ..., \xi_P \in \mathbb{S}^{d-1}$ such that

$$F(\|x\|) \approx \frac{1}{P}\sum_{p=1}^P f(|\langle \xi_p, x\rangle|) \quad \text{for all} \quad x \in \mathbb{R}^d. \tag{4}$$

Then, the kernel sums (1) read as

$$s_m = \sum_{n=1}^N w_n K(x_n, y_m) = \sum_{n=1}^N w_n F(\|x_n - y_m\|) \approx \frac{1}{P}\sum_{p=1}^P\sum_{n=1}^N w_n f(|\langle \xi_p, x_n - y_m\rangle|). \tag{5}$$

For computing the one-dimensional sums $\sum_{n=1}^{N} w_n f(|\langle \xi_p, x_n - y_m \rangle|)$ for all $m = 1, ..., M$, there exists algorithms with complexity $\mathcal{O}(M + N)$ or $\mathcal{O}((M + N) \log(M + N))$ in literature. These include fast summations based on non-equispaced Fourier transforms (Kunis et al., 2006; Potts et al., 2004), fast multipole methods (Greengard & Rokhlin, 1987) or sorting algorithms (Hertrich et al., 2024). In particular, we can approximate the vector $s = (s_1, ..., s_M)$ via (5) with a complexity of $\mathcal{O}(P(M + N))$.

## 2.2 Error Bounds for Uniformly Distributed Slices

To bound the error of the slicing procedure from the previous subsection, we consider error estimates for the approximation in (4). To this end, we assume that the directions $\xi_1, ..., \xi_P$ are iid samples from the uniform distribution on the sphere. Then, we exactly compute the variance

$$\mathbb{V}_d[f](x) := \mathbb{E}_{\xi \sim \mathcal{U}_{\mathbb{S}^{d-1}}} \left[ (f(|\langle \xi, x \rangle|) - F(\|x\|))^2 \right], \tag{6}$$

which bounds the mean squared error through the Bienaymé's identity as

$$\mathbb{E}_{\xi_1, ..., \xi_P \sim \mathcal{U}_{\mathbb{S}^{d-1}}} \left[ \left( \frac{1}{P} \sum_{p=1}^{P} f(|\langle \xi_p, x \rangle|) - F(\|x\|) \right)^2 \right] = \frac{\mathbb{V}_d[f](x)}{P}.$$

In particular, our results show relative error bounds of the negative distance kernel $K(x, y) = -\|x - y\|$ and the thin plate spline (except around $\|x\| = 1$), which do not depend on the dimension $d$. The proof is given in Appendix C.

**Theorem 1.** *Let* $F \colon \mathbb{R}_{\geq 0} \to \mathbb{R}$ *and* $f \colon \mathbb{R}_{\geq 0} \to \mathbb{R}$ *fulfill the slicing relation* (2).

i) *If* $F(\|\cdot\|)$ *is continuous and positive definite on* $\mathbb{R}^d$, *then* $\mathbb{V}_d[f](x) \leq F(0)^2 - F(\|x\|)^2$.

ii) *For the generalized Riesz kernel* $F(\|x\|) = -\|x\|^r$ *with* $r > 0$, *we have*

$$\mathbb{V}_d[f](x) = \left( \frac{\sqrt{\pi}\Gamma(r + \frac{1}{2})\Gamma(\frac{d+r}{2})^2}{\Gamma(\frac{r+1}{2})^2\Gamma(\frac{d}{2})\Gamma(r + \frac{d}{2})} - 1 \right) F(\|x\|)^2 < \frac{\sqrt{\pi}\Gamma(r + \frac{1}{2})}{\Gamma(\frac{r+1}{2})^2} F(\|x\|)^2. \tag{7}$$

iii) *For the thin plate spline* $F(\|x\|) = \|x\|^2 \log(\|x\|)$, *we have*

$$\mathbb{V}_d[f](x) = \left( 2 + \frac{c_1}{\log(\|x\|)} + \frac{c_2 + \mathcal{O}(d^{-1}\log d)}{\log(\|x\|)^2} \right) \left( 1 + \frac{2}{d} \right) F(\|x\|)^2,$$

*with* $c_1 \approx 4.189$ *and* $c_2 \approx 2.895$ *given in* (18).

iv) *For* $F(\|x\|) = \sum_{n=0}^{\infty} a_n \|x\|^n$ *and* $d \geq 3$ *odd, we have*

$$\mathbb{V}_d[f](x) = \sum_{n=0}^{\infty} \left( \sum_{k=0}^{n} \left( \prod_{i=1}^{(d-1)/2} \left( 1 + \frac{k(n-k)}{(2i-1)(n+2i-1)} \right) - 1 \right) a_k a_{n-k} \right) \|x\|^n.$$

*In particular, for the Laplace kernel* $F(\|x\|) = \exp(-\alpha\|x\|)$, *we have*

$$\mathbb{V}_3[f](x) = \frac{1}{4\alpha\|x\|} \left( 1 - (2(\alpha\|x\|)^2 + 2\alpha\|x\| + 1)F(\|x\|)^2 \right),$$

*and for the Gauss kernel* $F(\|x\|) = \exp(-\|x\|^2/(2\sigma^2))$, *we have*

$$\mathbb{V}_3[f](x) = \frac{\sigma^2}{2\|x\|^2} \left( 1 - \left( \frac{\|x\|^4}{2\sigma^4} + \frac{\|x\|^2}{\sigma^2} + 1 \right) F(\|x\|)^2 \right).$$

Some weaker error bounds for the Gauss and Riesz kernel were also shown in Hertrich (2024), see Appendix D.1 for a detailed comparison. In all cases, the variance is independent of the dimension $d$. The dependence on $x$ differs between the kernels: For positive definite kernels, which are always bounded, we have an absolute error bound i) independent of $\|x\|$. For the Riesz kernel, we have a relative error bound in ii). For the thin plate spline, iii) behaves like a relative bound for $\|x\| \to \infty$ and $\|x\| \to 0$, but as a constant around $\|x\| = 1$, which is a zero of $F$. For the Laplace and Gauss kernel, the dependence on $\|x\|$ changes drastically between $\|x\| \to \infty$ and $\|x\| \to 0$. In fact, $\mathbb{V}_3[f](x)$ is monotonically increasing in $\|x\|$ with global upper bound $1/(4\alpha)$, and converges quadratically in $\|x\|$ to zero for $\|x\| \to 0$. For the case $d > 3$, we conjecture a similar behavior, see Appendix D.2 for the discussion.

## 3 QUASI-MONTE CARLO SLICING

For directions drawn independently from the uniform measure $\mathcal{U}_{\mathbb{S}^{d-1}}$ on the sphere $\mathbb{S}^{d-1}$, our experiments from the numerical part suggest that the rate $\mathcal{O}(1/\sqrt{P})$ from Theorem 1 is optimal. As a remedy, we employ quadrature and quasi-Monte Carlo designs on the sphere for improving these error rates. To this end, we first revisit the corresponding literature in Subsection 3.1. Afterwards, we apply these results for our slicing method in Subsection 3.2.

### 3.1 QUASI-MONTE CARLO METHODS ON THE SPHERE

Let $\boldsymbol{\xi}^P = (\xi_1^P, ..., \xi_P^P) \in (\mathbb{S}^{d-1})^P$ for $P \in \mathbb{N}$. In the following, we aim to construct $\boldsymbol{\xi}^P$ such that the worst case error in a certain Sobolev space is asymptotically optimal. The definition of the Sobolev space $H^s(\mathbb{S}^{d-1})$ is given in Appendix B.

**Definition 2.** *A sequence $(\boldsymbol{\xi}^P)_P$ with $P \to \infty$ is called a sequence of QMC designs for $H^s(\mathbb{S}^{d-1})$ if there exists some $c(s,d) > 0$ independent of $P$ such that the worst case error*

$$\sup_{\substack{g \in H^s(\mathbb{S}^{d-1}) \\ \|g\|_{H^s(\mathbb{S}^{d-1})} \leq 1}} \left| \frac{1}{|\mathbb{S}^{d-1}|} \int_{\mathbb{S}^{d-1}} g(\xi) \mathrm{d}\xi - \frac{1}{P} \sum_{p=1}^{P} g(\xi_p^P) \right| \leq \frac{c(s,d)}{P^{s/(d-1)}} \in \mathcal{O}(P^{-s/(d-1)}). \quad (8)$$

It was proven by Hesse (2006) that the error rate $\mathcal{O}(P^{-s/(d-1)})$ is optimal, see also Brauchart et al. (2014). For $s > \frac{d-1}{2}$, the existence of sequences of QMC designs is ensured by so-called spherical designs. More precisely, $\boldsymbol{\xi}^P$ is called a spherical $t$-design if the quadrature at these points integrates all polynomials of degree $t \in \mathbb{N}$ exactly, i.e., if it holds

$$\frac{1}{|\mathbb{S}^{d-1}|} \int_{\mathbb{S}^{d-1}} \psi(\xi) \mathrm{d}\xi = \frac{1}{P} \sum_{p=1}^{P} \psi(\xi_p^P) \quad \text{for all polynomials } \psi \text{ of degree } \leq t.$$

It can be shown that for any $t$ there exists a spherical $t$-design with $P = \mathcal{O}(t^{d-1})$ points, see Bondarenko et al. (2013). By (Brauchart & Hesse, 2007, Cor 3.6), such a sequence of spherical $t$-designs is a sequence of QMC designs, see also (Brauchart et al., 2014, Sect 1) for a summary.

Unfortunately, the construction of spherical $t$-designs in arbitrary dimension is numerically intractable. Instead, many QMC methods rely on low-discrepancy point sets. It was shown in (Brauchart et al., 2014, Thm 14) that a sequence $\boldsymbol{\xi}^P$ that minimizes the sum of powers of Euclidean distances

$$\mathcal{E}(\boldsymbol{\xi}^P) := - \sum_{p,q=1}^{P} \|\xi_p^P - \xi_q^P\|^{2s-d-1} \quad (9)$$

is a QMC design for $H^s(\mathbb{S}^{d-1})$ for $s \in (\frac{d-1}{2}, \frac{d+1}{2})$. In the numerics, we will consider $s = \frac{d}{2}$, so that we get a QMC design for $H^{d/2}(\mathbb{S}^{d-1})$, which we call the distance QMC design. Note that, up to a constant, (9) coincides with the maximum mean discrepancy with the Riesz kernel $K(x,y) = -\|x - y\|^{2s-d-1}$ between the probability measures $\frac{1}{P} \sum_{i=1}^{P} \delta_{\xi_p^P}$ and $\mathcal{U}_{\mathbb{S}^{d-1}}$, which is also known as energy distance Székely (2002). Furthermore, one can easily transform a QMC sequence into an unbiased estimator in (4), see Appendix E.

### 3.2 SMOOTHNESS OF ONE-DIMENSIONAL BASIS FUNCTIONS

In order to apply the above theorems for our approximation (2), we need to ensure that for any $x \in \mathbb{R}^d$ the spherical function $g_x : \mathbb{S}^{d-1} \to \mathbb{R}$ given by

$$g_x(\xi) := f(|\langle \xi, x \rangle|) \quad (10)$$

is sufficiently smooth on $\mathbb{S}^{d-1}$. For some specific examples, this is verified in the next theorem, whose proof is given in Appendix F. For part ii), we explicitly compute the Sobolev norm of $g_x$. Note that, in the special case $s = 0$, this relates to the variance (6) by the formula $\|g_x\|_{H^0(\mathbb{S}^{d-1})}^2 = |\mathbb{S}^{d-1}|(\mathbb{V}_d[f](x) + F(\|x\|)^2)$ for $(f, F)$ fulfilling (2).

**Theorem 3.** *Let $x \in \mathbb{R}^d$ with $x \neq 0$. For the Gauss, Riesz and Matérn kernel, the following smoothness results hold true:*

i) *For $F(t) = \exp(-\frac{t^2}{2\sigma^2})$, we have $g_x \in H^s(\mathbb{S}^{d-1})$ for all $s \geq 0$.*

ii) *For $F(t) = t^r$ with $t \geq 0$ and $r > -1$, we have $g_x \in H^s(\mathbb{S}^{d-1})$ if and only if $s < r + \frac{1}{2}$.*

iii) *For $F(t) = \frac{2^{1-\nu}}{\Gamma(\nu)}(\frac{\sqrt{2\nu}}{\beta}t)^\nu K_\nu(\frac{\sqrt{2\nu}}{\beta}t)$, $t \geq 0$, we have $g_x \in H^s(\mathbb{S}^{d-1})$ if $s < 2\nu + \frac{1}{2}$.*

Note that the theorem also includes the Laplace kernel, which is the Matérn kernel for $\nu = \frac{1}{2}$. Combining this theorem with the results from the previous subsection leads to improved error bounds for the Gauss and Matérn kernel in the following corollary. For the Riesz kernel, the last theorem can be seen as a negative result that the theory from the previous subsection is not applicable.

**Corollary 4.** *Let $d \in \mathbb{N}$ and $s > \frac{d-1}{2}$. Then there exists a constant $c(s,d)$ and a sequence $(\boldsymbol{\xi}^P)_P$ with $P \to \infty$ such that for the Gauss and Matérn kernel with basis functions $F(t) = \exp(-\frac{t^2}{2\sigma^2})$ and $F(t) = \frac{2^{1-\nu}}{\Gamma(\nu)}(\frac{\sqrt{2\nu}}{\beta}x)^\nu K_\nu(\frac{\sqrt{2\nu}}{\beta}t)$ with $\nu > \frac{2s-1}{4}$, respectively, it holds that*

$$\sup_{x \in \mathbb{R}^d} \left| F(\|x\|) - \frac{1}{P}\sum_{p=1}^{P} f(|\langle \xi_p, x \rangle|) \right| \leq \frac{c(s,d)}{P^{\frac{s}{d-1}}}.$$

*For $s \in (\frac{d-1}{2}, \frac{d+1}{2})$, such $\boldsymbol{\xi}^P$ are given by minimizers of (9).*

In Appendix J, we derive the complexity of the slicing-based kernel summation with a QMC sequence and show in Proposition 5 that it is superior to the random Fourier feature approach for the Gauss kernel. The summation methods are described in Appendix I and Appendix G, respectively.

## 4 NUMERICAL EXAMPLES

In the following, we evaluate the kernel approximation with QMC slicing for several QMC sequences and compare our results with random Fourier feature-based (RFF, Rahimi & Recht, 2007) methods. We implement the comparison in Julia and Python, and provide the code online[3]. Additionally, we provide a general PyTorch implementation for fast kernel summations via slicing and non-equispaced fast Fourier transforms[4]. In Subsection 4.1, we describe the used QMC sequences, RFF-methods and kernels. Afterwards, in Subsection 4.2, we numerically evaluate the approximation error in (4). Finally, we apply our approximation for fast kernel summations in Subsection 4.3. We include additional numerical examples in Appendix K.

### 4.1 QMC SEQUENCES AND KERNELS

**QMC Sequences** Beside standard slicing where the projections $\boldsymbol{\xi}^P$ are drawn iid from the uniform distribution on $\mathbb{S}^{d-1}$, we consider the following sequences. We would like to emphasize that for the first two of them it is not clear, whether they are QMC designs in the sense of Definition 2, even though they are sometimes called QMC sequences in the literature.

- **Sobol Sequence**: Two commonly used QMC sequences on $[0,1]^d$ are Sobol (Sobol', 1967) and Halton (Halton, 1960) sequences. They can be transformed to QMC sequences for the normal distribution by applying the inverse cumulative density function along each dimension of the sequence, which was used for deriving a QMC method for random Fourier features Avron et al. (2016). To obtain a potential QMC sequence on the sphere, Nguyen et al. (2024) proposed to project the QMC sequence for the multivariate normal distribution onto the sphere by the transformation $\xi = \theta/\|\theta\|$, see also Beltrán et al. (2023). It is not known whether this constitutes a QMC sequence in the sense of Definition 2. To generate the original Sobol sequence on $[0,1]^d$, we use the implementation from `SciPy` (Virtanen et al., 2020) in Python and the `Sobol.jl` package in Julia. Our numerical experiments

---

[3]available at `https://github.com/johertrich/fastsum_QMC_slicing`
[4]available at `https://github.com/johertrich/simple_torch_NFFT`

suggest that the Sobol sequence lead to slightly better results than the Halton sequence. Therefore, we omit the Halton sequence in our comparison.

- **Orthogonal:** Even though this is technically not a QMC sequence, we adapt the approach of orthogonal features (Yu et al., 2016) for slicing and generate directions $\xi$ as follows: We generate $\lceil \frac{P}{d} \rceil$ orthogonal matrices from the uniform distribution on $O(d)$ (this can be done by taking the Q-factor of the QR decomposition applied on a matrix with standard normally distributed entries). Together, these matrices have $d\lceil \frac{P}{d} \rceil$ columns from which we choose $\boldsymbol{\xi}^P$ to be the first $P$ of those.

- **Distance:** In Section 3.1, we considered the distance QMC design $\boldsymbol{\xi}^P$ for $H^{\frac{d}{2}}(\mathbb{S}^{d-1})$, which is a minimizer of $\mathcal{E}(\boldsymbol{\xi}^P) = -\sum_{p,q=1}^P \|\xi_p^P - \xi_q^P\|$, see (9). In our application, we have the additional symmetry that $f(|\langle x, \xi \rangle|) = f(|\langle x, -\xi \rangle|)$. Therefore, we construct the distance QMC designs $\boldsymbol{\xi}^P$ by minimizing the functional $\mathcal{E}_{\text{sym}}(\boldsymbol{\xi}^P) := \mathcal{E}((\boldsymbol{\xi}^P, -\boldsymbol{\xi}^P))$. We do this numerically with the Adam optimizer (Kingma & Ba, 2015) and the PyKeops package (Charlier et al., 2021), which takes from a couple of seconds (for $d = 3$) up to one hour (for $d = 200$ and $P \approx 5000$) on an NVIDIA RTX 4090 GPU. In Appendix H, we show that if $P \leq d$, the orthogonal points from above minimize $\mathcal{E}_{\text{sym}}$, so this approach differs only if $P > d$.

- **Spherical Design**: For $d = 3$, several spherical $t$-designs on the $\mathbb{S}^2$ were computed by Gräf & Potts (2011) up to $t \leq 1000$ and $P \leq 1002000$ and are available online[5]. Spherical $t$-designs for $\mathbb{S}^3$ were computed by Womersley (2018) up to $t \leq 31$ and $P \leq 3642$. Unfortunately, the computation in higher dimensions appears to be intractable such that we only use the spherical designs for $d = 3$.

**Compared Methods**   We compare our results with the following methods:

- **Random Fourier Features** (RFF, Rahimi & Recht, 2007): see Appendix G for a description.

- **Orthogonal Random Features** (ORF, Yu et al., 2016): The directions of the RFF features are chosen in the same way as explained above for the orthogonal slicing.

- **QMC-Random Fourier Features** (Sobol RFF, Avron et al., 2016): For the Gauss kernel, we also compare with QMC random Fourier features, which are only applicable for kernels where the Fourier transform decouples as a product over the dimensions. For the kernels from Table 2, this is only true for the Gauss kernel. As a QMC sequence in $[0, 1]^d$, we choose the Sobol sequence.

**Kernels**   We use the Gauss, Laplace, Matérn (with $\nu = p + \frac{1}{2}$ for $p \in \{1, 3\}$), the negative distance kernel (Riesz kernel with $r = 1$) and the thin plate spline kernel, see Table 2 in the appendix for the pairs $(f, F)$. The parameters $\sigma$, $\alpha$ and $\beta$ are chosen by the median rule (see, e.g., Garreau et al., 2017 for an overview and history). That is, we choose $\sigma = \beta = \frac{1}{\alpha} = \gamma m$, where $m$ is the median of all considered input norms $\|x\|$ of the basis functions $F$ and $\gamma$ is some scaling factor which we set to $\gamma \in \{\frac{1}{2}, 1, 2\}$.

## 4.2   NUMERICAL EVALUATION OF THE SLICING ERROR

We examine the approximation error in (4) numerically. To this end, we draw a sample $x$ from $\mathcal{N}(0, 0.1I)$ and evaluate the absolute error $\left| F(\|x\|) - \frac{1}{P}\sum_{p=1}^P f(|\langle \xi_p, x \rangle|) \right|$. We average this error over 50 realizations of $\boldsymbol{\xi}^P$ (whenever $\boldsymbol{\xi}^P$ is random) and 1000 samples of $x$. The average results for the scale factor $\gamma = 1$ and dimension $d \in \{3, 10, 50\}$ with the Gauss, Laplace and Matérn kernel are given in Figure 1. We observe that all methods despite the spherical designs for the Gauss kernel converge with rate $\mathcal{O}(P^{-r})$ for some $r > 0$. To estimate the rate $r$ numerically, we fit a regression line in the loglog plot. The resulting rates $r$ are given in Table 1. Further plots and tables are given in Appendix K.1 considering the negative distance kernel, higher dimensions and smaller/larger kernel widths.

Overall, the distance QMC designs perform best in most examples, except when the (provably optimal) spherical designs are applicable, which are only computable in $d \leq 4$ and outperform the

---

[5] https://www-user.tu-chemnitz.de/~potts/workgroup/graef/quadrature/index.php.en

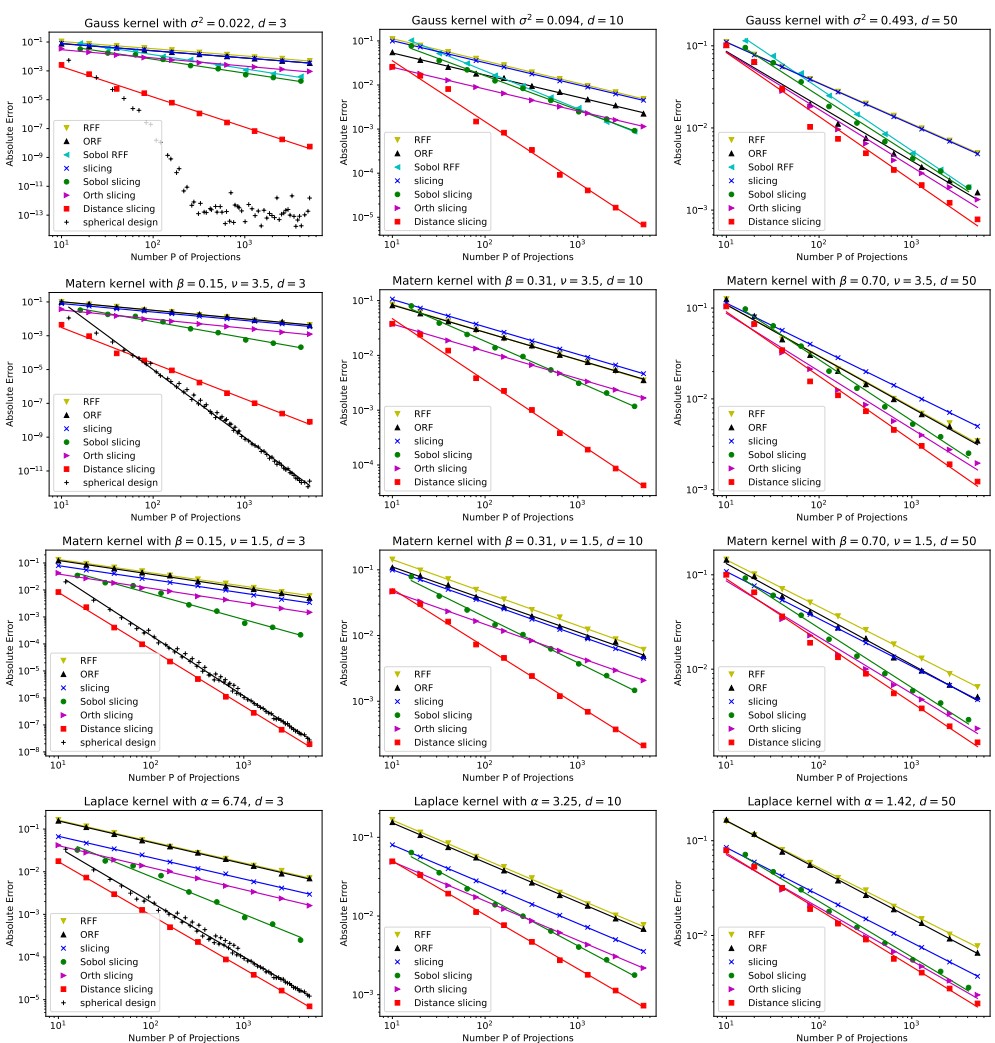

Figure 1: Loglog plot of the approximation error $\left| F(\|x\|) - \frac{1}{P}\sum_p f(|\langle \xi_p, x\rangle|)\right|$ for approximating the function $F$ by slicing (4) versus the number $P$ of projections (or the number $D = P$ of features for RFF and ORF) for different kernels and dimensions (left $d = 3$, middle $d = 10$, right $d = 50$). The results are averaged over 50 realizations of $\boldsymbol{\xi}^P$ and 1000 realizations of $x$. The kernel parameters are set by the median rule with scaling factor $\gamma = 1$. We fit a regression line in the loglog plot for each method to estimate the convergence rate $r$, see also Table 1.

Table 1: Estimated convergence rates for the different methods. We estimate the rate $r$ by fitting a regression line in the loglog plot. Then, we obtain the estimated convergence rate $P^{-r}$ for some $r > 0$. Consequently, larger values of $r$ correspond to a faster convergence. The resulting values of $r$ are given in the below tables, the best values are highlighted in bold. The kernel parameters are the same as in Figure 1 (median rule with scaling factor $\gamma = 1$).

| Gauss kernel with median rule and scaling $\gamma = 1$ | | | | | | | |
|---|---|---|---|---|---|---|---|
| | RFF-based | | | Slicing-based | | | |
| Dimension | RFF | Sobol | ORF | Slicing | Sobol | Orth | Distance |
| $d = 3$ | 0.50 | 0.98 | 0.50 | 0.50 | 0.96 | 0.57 | **2.10** |
| $d = 10$ | 0.50 | 0.86 | 0.50 | 0.50 | 0.78 | 0.50 | **1.38** |
| $d = 50$ | 0.50 | 0.76 | 0.67 | 0.50 | 0.72 | 0.70 | **0.78** |

| Matérn kernel with $\nu = 3 + \frac{1}{2}$ and median rule with scaling $\gamma = 1$ | | | | | | | |
|---|---|---|---|---|---|---|---|
| | RFF-based | | Slicing-based | | | | |
| Dimension | RFF | ORF | Slicing | Sobol | Orth | Distance | spherical design |
| $d = 3$ | 0.51 | 0.51 | 0.50 | 0.96 | 0.54 | 2.11 | **4.01** |
| $d = 10$ | 0.51 | 0.50 | 0.50 | 0.74 | 0.50 | **1.13** | - |
| $d = 50$ | 0.56 | 0.57 | 0.50 | 0.67 | 0.64 | **0.71** | - |

| Matérn kernel with $\nu = 1 + \frac{1}{2}$ and median rule with scaling $\gamma = 1$ | | | | | | | |
|---|---|---|---|---|---|---|---|
| | RFF-based | | Slicing-based | | | | |
| Dimension | RFF | ORF | Slicing | Sobol | Orth | Distance | spherical design |
| $d = 3$ | 0.50 | 0.51 | 0.51 | 0.95 | 0.53 | 2.11 | **2.24** |
| $d = 10$ | 0.50 | 0.50 | 0.50 | 0.70 | 0.50 | **0.89** | - |
| $d = 50$ | 0.50 | 0.54 | 0.50 | 0.63 | 0.60 | **0.66** | - |

| Laplace kernel with median rule and scaling $\gamma = 1$ | | | | | | | |
|---|---|---|---|---|---|---|---|
| | RFF-based | | Slicing-based | | | | |
| Dimension | RFF | ORF | Slicing | Sobol | Orth | Distance | spherical design |
| $d = 3$ | 0.50 | 0.50 | 0.50 | 0.88 | 0.52 | 1.26 | **1.28** |
| $d = 10$ | 0.50 | 0.50 | 0.50 | 0.63 | 0.50 | **0.68** | - |
| $d = 50$ | 0.49 | 0.52 | 0.50 | 0.59 | 0.56 | **0.60** | - |

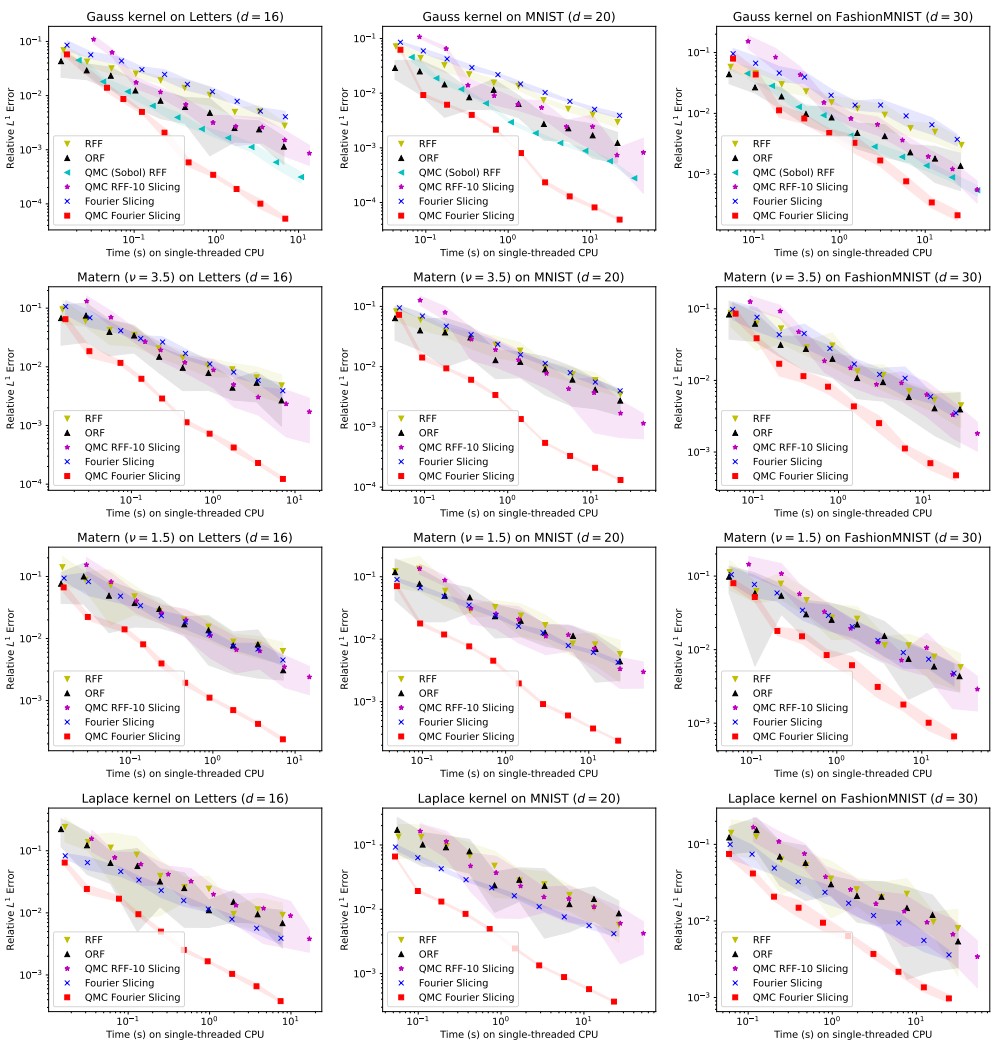

Figure 2: Loglog plot of the relative $L^1$ approximation error versus computation time for computing the kernel summations (1) with different kernels and methods. We use the Letters dataset ($M = N = 20000$ points), MNIST (reduced to dimension $d = 20$ via PCA, $M = N = 60000$ points) and FashionMNIST (reduced to dimension $d = 30$ via PCA, $M = N = 60000$ points). We run each method 10 times. The shaded area indicates the standard deviation of the error. For Fourier slicing, we use $P = 5 \cdot 2^k$ slices for $k = 1, ..., 10$. In order to obtain similar computation times, we use $5 \cdot 2^{k-1}$ slices for RFF-10 slicing and $D = 2P$ features for RFF and ORF.

distance QMC designs for smooth kernels as they reach machine precision already for $P \approx 250$. In accordance with Corollary 4, the benefits of QMC slicing are better for smooth kernels and in low dimensions. But also for $d = 50$, a significant advantage of QMC slicing is visible. In particular, we often observe much faster convergence rates than the proven worst case error rates $r = \frac{d}{2(d-1)}$ on $H^s(\mathbb{S}^{d-1})$ with the distance QMC designs, see Section 3. Furthermore, the slight advantage of the distance QMC designs versus the spherical designs for the Laplace kernel is because the former ones are chosen specifically for symmetric functions.

## 4.3 FAST KERNEL SUMMATION

Finally, we test our kernel approximation for computing the whole kernel sums from (1). For computing the one-dimensional kernel sums $\sum_{n=1}^{N} w_n f(|\langle \xi_p, x_n - y_m \rangle|)$, we use the following methods combined with either random or QMC points on the sphere:

- **(QMC) Sorting-Slicing:** For the negative distance kernel, we use the sorting algorithm from Hertrich et al. (2024), see also (Hertrich, 2024, Sec 3.2).

- **(QMC) Fourier-Slicing:** For the Gauss, Matérn and Laplace kernel, we use fast Fourier summation based on the non-equispaced fast Fourier transform (NFFT) for the one-dimensional kernel summation. A general overview on NFFTs and fast Fourier summation can be found in the text book (Plonka et al., 2023, Sec 7.5). Similar as in (Hertrich, 2024, Sec 2.3), we do not evaluate the one-dimensional basis functions, but directly compute the Fourier transforms. We revisit the background on the one-dimensional fast Fourier summation and specify the used parameters in Appendix I.

- **(QMC) RFF-$k$ Slicing:** For positive definite kernels (Gauss, Laplace, Matérn), we use one-dimensional random Fourier features with $k$ features for the basis function $f$. For iid or orthogonal slices and $k = 1$, this approach is related to RFF and ORF, as outlined in Appendix G.

For $\boldsymbol{\xi}^P$, we use the distance QMC designs, since we have seen in the previous subsection that it performs best among the QMC rules. Moreover, we use the randomization from Appendix E for the QMC design to obtain an unbiased estimator. We evaluate the kernel sums on Letters dataset ($d = 16$, Slate, 1991), MNIST (reduced to $d = 20$ dimensions via PCA, LeCun et al., 1998) and FashionMNIST (reduced to $d = 30$ dimensions via PCA, Xiao et al., 2017), where $(x_1, ..., x_N)$ and $(y_1, ..., y_M)$ constitute the whole dataset and the weights $(w_1, ..., w_N)$ are set to 1. In particular, we have $M = N = 20000$ for the Letters dataset and $M = N = 60000$ for MNIST and Fashion-MNIST. Then, we approximate the vector $s = (s_1, ..., s_M)$ from (1) and report the absolute error $\|s - s_{\text{true}}\|_1$. We choose the kernel parameters by the median rule with scale factor $\gamma = 1$ based on 1000 example pairs $(x, y)$. We benchmark the computation times on a single thread of an AMD Ryzen Threadripper 7960X CPU and compare our results with RFF, ORF and (non-QMC) slicing. Since the QMC designs $\boldsymbol{\xi}^P$ depend neither on the dataset nor on the kernel, we consider its construction not as a part of the computation time. For the Gauss kernel, we also compare with QMC (Sobol) RFF (Avron et al., 2016), which is not applicable for the other kernels. We use $P = 5 \cdot 2^k$ slices for $k = 1, ..., 10$ in the slicing method. In order to obtain similar computation times, we use $5 \cdot 2^{k-1}$ slices for RFF-10 slicing and $D = 2P$ features for RFF and ORF.

We visualize the approximation error (including standard deviations) in Figure 2 for the Gauss, Laplace and Matérn kernel. The results for the negative distance kernel, for the thin plate spline kernel, for higher dimensional datasets (including the full MNIST and FashionMNIST with $d = 784$) and a GPU comparison are included in Appendix K.2. In addition, we apply the fast kernel summation to computing MMD gradient flows in Appendix K.3. We can see that QMC Fourier-slicing has a significantly smaller error than the other methods. Moreover, it has the smallest standard deviation of the error.

## 5 CONCLUSIONS

**Summary and Outlook** We proposed a slicing approach to compute large kernel sums in $\mathcal{O}(P(N + M + N_{\text{ft}} \log N_{\text{ft}}))$ instead of the naïve $\mathcal{O}(NM)$ arithmetic operations. In the case of iid directions, we proved error bounds with rate $\mathcal{O}(1/\sqrt{P})$. To improve this rate, we proposed a QMC approach based on spherical quadrature rules. We demonstrated by numerical methods that our QMC slicing approach outperforms existing methods, where the advantage is most significant for dimensions $d \leq 100$. In the future, we want to improve our theoretical analysis on QMC slicing in order to match the convergence rate from the numerical section. One possible approach for that could be to study worst case errors for symmetric functions on the sphere, since the mappings $\xi \to g_x(\xi)$ from Section 3.2 are always symmetric, see Appendix K.4 for details. From a practical side, we want to apply the slicing approach in some actual applications.

**Limitations** In the numerical part, we observe significantly better error rates than we can prove theoretically, see Appendix K.4 for a discussion. Moreover, the computation of the QMC directions can be very costly and depends strongly on the chosen method. For the spherical designs, it is even intractable for high dimensions. Finally, the advantage of QMC slicing becomes smaller for higher dimensions, which is a well-known effect for most QMC methods.

ACKNOWLEDGMENTS

JH acknowledges funding by the German Research Foundation (DFG) within the Walter Benjamin Programme with project number 530824055 and by the EPSRC programme grant *The Mathematics of Deep Learning* with reference EP/V026259/1. TJ acknowledges funding by the Deutsche Forschungsgemeinschaft under Germany's Excellence Strategy EXC-2046/1, Projekt-ID 390685689 (The Berlin Mathematics Research Center MATH+). MQ acknowledges funding by the German Research Foundation (DFG) with reference STE 571/19-2 and project number 495365311, within the Austrian Science Fund (FWF) SFB 10.55776/F68 *Tomography Across the Scales*. For open access purposes, the authors have applied a CC BY public copyright license to any author-accepted manuscript version arising from this submission. We gratefully thank Gabriele Steidl for fruitful conversations.

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

## A  BASIS FUNCTION PAIRS $(F, f)$

We include a list of pairs of basis functions $(F, f)$ that fulfill the slicing formula and the corresponding Fourier transforms in Table 2. The pairs are taken from (Hertrich, 2024, Table 1). We use the convention that the Fourier transform of a function $g \in L_1(\mathbb{R}^d)$ is defined by

$$\mathcal{F}_d[g](\omega) = \int_{\mathbb{R}^d} g(x) e^{-2\pi i \langle \omega, x \rangle} dx. \tag{11}$$

The basis functions from the table involve a couple of special functions, which are defined as follows:

- Gamma function: $\Gamma(z) = \int_0^\infty t^{z-1} e^{-t} dt$, $\mathrm{Re}(z) > 0$,

- Modified Bessel function of first kind: $I_\alpha(x) = \sum_{m=0}^\infty \frac{1}{m! \Gamma(m+\alpha+1)} \left(\frac{x}{2}\right)^{2m+\alpha}$,

- Modified Bessel function of second kind: $K_\alpha(x) = \frac{\pi}{2} \frac{I_{-\alpha}(x) - I_\alpha(x)}{\sin(\alpha\pi)}$,

- Modified Struve function: $\mathbf{L}_\alpha(x) = \sum_{m=0}^\infty \frac{1}{\Gamma(m+\frac{3}{2})\Gamma(m+\alpha+\frac{3}{2})} \left(\frac{x}{2}\right)^{2m+\alpha+1}$.

The formula for the Fourier transform of the Matérn kernel (and thus for the Laplace kernel with $\nu = 1/2$) can be found in (Williams & Rasmussen, 2006, (4.15)). Consequently, we recover the well-known results that the Fourier transforms of the Gauss, Laplace and Matérn kernels are the Fourier transforms of the Gauss, Cauchy and Student-$t$ (with $2\nu$ degrees of freedom) distribution. To compute the Fourier transforms $\mathcal{F}_1^{-1}[f(|\cdot|)]$, we apply (Rux et al., 2025, Prop 3.1), see also (Hertrich, 2024, Lem 6) for the Gauss kernel. For the Riesz and thin plate spline kernel, the Fourier transform does not exist in a classical sense, but only in a distributional one, see Rux et al. (2025) and (Wendland, 2004, Sect 8.3) for details. For the sliced Laplace $f(x) = \exp(-\alpha x)$, we have by (Hertrich, 2024, (3)) and (Gradshteyn & Ryzhik, 2007, 3.387.5)

$$F(t) = \int_0^1 \exp(-\alpha s t)(1-s^2)^{\frac{d-3}{2}} ds = \frac{\sqrt{\pi} 2^{\frac{d-4}{2}} \Gamma(\frac{d-1}{2})}{(\alpha t)^{\frac{d-2}{2}}} \left( I_{\frac{d-2}{2}}(-\alpha t) + \mathbf{L}_{\frac{d-2}{2}}(-\alpha t) \right). \tag{12}$$

Table 2: Basis functions $F$ for different kernels $K(x,y) = F(\|x - y\|)$ and corresponding basis functions $f$ from $\mathrm{k}(x,y) = f(|x-y|)$. We added the inverse Fourier transforms $\mathcal{F}_d^{-1}[F(\|\cdot\|)]$ and $\mathcal{F}_1^{-1}[f(|\cdot|)]$ to the table.

| Kernel | $F(x)$ | $\mathcal{F}_d^{-1}[F(\|\cdot\|)](\|\omega\|)$ | $f(x)$ | $\mathcal{F}_1^{-1}[f](\|\omega\|)$ |
|---|---|---|---|---|
| Gauss | $\exp(-\frac{x^2}{2\sigma^2})$ | $(2\pi\sigma^2)^{d/2}\exp(-2\pi^2\sigma^2\omega^2)$ | ${}_1F_1(\frac{d}{2};\frac{1}{2};\frac{-x^2}{2\sigma^2})$ | $\frac{\pi\sigma\exp(-2\pi^2\sigma^2\omega^2)(2\pi^2\sigma^2\omega^2)^{(d-1)/2}}{2\Gamma(\frac{d}{2})}$ |
| Laplace | $\exp(-\alpha x)$ | $\frac{\Gamma(\frac{d+1}{2})2^d \pi^{\frac{d-1}{2}}}{\alpha^d}(1+\frac{4\pi^2\omega^2}{\alpha^2})^{-\frac{d+1}{2}}$ | $\sum_{n=0}^\infty \frac{(-1)^n \alpha^n \sqrt{\pi}\Gamma(\frac{n+d}{2})}{n!\Gamma(\frac{d}{2})\Gamma(\frac{n+1}{2})}x^n$ | $\frac{\Gamma(\frac{d+1}{2})2^d \pi^{\frac{d-1}{2}}|\omega|^{d-1}}{\Gamma(\frac{d}{2})\alpha^d}(1+\frac{4\pi^2\omega^2}{\alpha^2})^{-\frac{d+1}{2}}$ |
| Sliced Laplace | See (12) | $\frac{\Gamma(\frac{d}{2})}{\pi^{d/2}|\omega|^{d-1}}\frac{2\alpha}{\alpha^2+4\pi^2\omega^2}$ | $\exp(-\alpha x)$ | $\frac{2\alpha}{\alpha^2+4\pi^2\omega^2}$ |
| Matérn | $\frac{2^{1-\nu}}{\Gamma(\nu)}(\frac{\sqrt{2\nu}}{\beta}x)^\nu K_\nu(\frac{\sqrt{2\nu}}{\beta}x)$ | $\frac{\Gamma(\frac{2\nu+d}{2})2^{\frac{d}{2}}\pi^{\frac{d}{2}}\beta^d}{\Gamma(\nu)\nu^{\frac{d}{2}}}(1+\frac{2\pi^2\beta^2\omega^2}{\nu})^{-\frac{2\nu+d}{2}}$ | (Hertrich, 2024, Appendix C) | $\frac{\Gamma(\frac{2\nu+d}{2})2^{\frac{d}{2}}\pi^d\beta^d|\omega|^{d-1}}{\Gamma(\frac{d}{2})\Gamma(\nu)\nu^{\frac{d}{2}}}(1+\frac{2\pi^2\beta^2\omega^2}{\nu})^{-\frac{2\nu+d}{2}}$ |
| Riesz for $r > -1$ | $-x^r$ | not a function | $-\frac{\sqrt{\pi}\Gamma(\frac{d+r}{2})}{\Gamma(\frac{d}{2})\Gamma(\frac{r+1}{2})}x^r$ | not a function |
| Thin Plate Spline | $x^2\log(x)$ | not a function | $dx^2\log(x)+C_dx^2$, with $C_d=\frac{d}{2}(H_{\frac{d}{2}}-2+\log(4))$ | not a function |

## B  SPHERICAL SOBOLEV SPACES

We briefly introduce spherical harmonics and Sobolev spaces following Atkinson & Han (2012). We denote by $|\mathbb{S}^{d-1}| = \frac{2\pi^{d/2}}{\Gamma(d/2)}$ the volume of the unit sphere $\mathbb{S}^{d-1}$. Let $Y_n^k$, $k = 1, \ldots, N_{n,d}$ be an $L_2$-orthonormal basis of spherical harmonics of degree $n \in \mathbb{N}_0$, i.e., harmonic polynomials in $d$ variables that are homogeneous of degree $n$. Here, the dimension of the space of spherical harmonics of degree $n$ is

$$N_{n,d} = \frac{(2n+d-2)(n+d-3)!}{n!(d-2)!} \simeq \frac{2}{(d-2)!}n^{d-2} \quad \text{for} \quad n \to \infty. \tag{13}$$

The spherical harmonics $Y_n^k$, $n \in \mathbb{N}_0$, $k = 1, \ldots, N_{n,d}$ form an orthonormal basis of $L_2(\mathbb{S}^{d-1})$.

The spherical Sobolev space $H^s(\mathbb{S}^{d-1})$ for $s \in \mathbb{R}$ can be defined as the completion of $C^\infty(\mathbb{S}^{d-1})$ with respect to the norm

$$\|g\|^2_{H^s(\mathbb{S}^{d-1})} = \sum_{n=0}^\infty \sum_{k=1}^{N_{n,d}} \left(n + \tfrac{d-2}{2}\right)^{2s} |\langle g, Y_n^k \rangle_{L_2(\mathbb{S}^{d-1})}|^2, \tag{14}$$

where $\langle g, Y_n^k \rangle_{L_2(\mathbb{S}^{d-1})} = \int_{\mathbb{S}^{d-1}} g(\xi) Y_n^k(\xi) \mathrm{d}\xi$. Note that the factor $(n + \tfrac{d-2}{2})^{2d}$ can be replaced by another one with the same asymptotic behavior with respect to $n$, yielding an equivalent norm. For $s = 0$, we can identify $H^s(\mathbb{S}^{d-1})$ with $L_2(\mathbb{S}^{d-1})$. The Sobolev spaces are nested in the sense that $H^s(\mathbb{S}^{d-1}) \subset H^t(\mathbb{S}^{d-1})$ whenever $s > t$. If $s > \tfrac{d-1}{2}$, each function in the Sobolev space $H^s(\mathbb{S}^{d-1})$ is continuous (more specifically, it has a continuous representative). If $s$ is an integer, then $H^s(\mathbb{S}^{d-1})$ consists of all functions whose (distributional) derivatives up to order $s$ are square integrable, cf. (Dai & Xu, 2013, Sect 4.5 and 4.8). An alternative characterization of the Sobolev norm uses the Laplace–Beltrami operator $\Delta_*$, which consists of the spherical part of the Laplace, and is given by

$$\|g\|_{H^s(\mathbb{S}^{d-1})} = \left\| \left(-\Delta_* + (\tfrac{d-2}{2})^2\right)^{s/2} g \right\|_{L^2(\mathbb{S}^{d-1})}.$$

If $s$ is an even integer, the operator applied to $g$ is a usual differentiable operator, otherwise it is a pseudodifferential operator.

## C  PROOF OF THEOREM 1

**i):** Since $F$ is continuous and positive definite, so is $f$ by (Rux et al., 2025, Corollary 4.11). Further, because $\mathbb{E}[f(|\langle \xi, x \rangle|)] = F(\|x\|)$ and due to the fact that positive definite functions are maximal in the origin, we deduce

$$\mathbb{V}_d[f] = \mathbb{E}_{\xi \sim \mathcal{U}_{\mathbb{S}^{d-1}}}[f(|\langle \xi, x \rangle|)^2] - F(\|x\|)^2 \le f(0)^2 - F(\|x\|)^2.$$

**ii):** We write $\boldsymbol{\xi} = (\xi_1, ..., \xi_d)^T$, where $\xi_d \in [-1, 1]$ denotes the $d$-th component of $\boldsymbol{\xi}$. We assume w.l.o.g. that $x = \lambda e_d$ with $\lambda \ne 0$ and recall that we can decompose the unnormalized surface measure on $\mathbb{S}^{d-1}$ as

$$\mathrm{d}\mathbb{S}^{d-1}(\boldsymbol{\xi}) = \mathrm{d}\mathbb{S}^{d-2}(\boldsymbol{\eta})(1 - t^2)^{\frac{d-3}{2}} \mathrm{d}t,$$

where $\boldsymbol{\xi} = \sqrt{1 - t^2}\boldsymbol{\eta} + t e_d$ and $\boldsymbol{\eta} \in \mathbb{S}^{d-2} \times \{0\}$, see (Atkinson & Han, 2012, (1.16)). Consequently,

$$\mathbb{E}_{\boldsymbol{\xi} \sim \mathcal{U}_{\mathbb{S}^{d-1}}}\left[|\langle \boldsymbol{\xi}, x \rangle|^{2r}\right] = \frac{\|x\|^{2r}}{|\mathbb{S}^{d-1}|} \int_{\mathbb{S}^{d-1}} |\xi_d|^{2r} \mathrm{d}\mathbb{S}^{d-1}(\boldsymbol{\xi})$$

$$= \frac{\|x\|^{2r}}{|\mathbb{S}^{d-1}|} \int_{\mathbb{S}^{d-2}} \int_{-1}^1 |t|^{2r}(1-t^2)^{\frac{d-3}{2}} \mathrm{d}\mathbb{S}^{d-2}(\boldsymbol{\eta}) \mathrm{d}t = \frac{\|x\|^{2r}}{|\mathbb{S}^{d-1}|}|\mathbb{S}^{d-2}| \int_{-1}^1 (t^2)^r (1-t^2)^{\frac{d-3}{2}} \mathrm{d}t.$$

Further, with $B(\cdot, \cdot)$ denoting the Beta function and the substitution $u = t^2$, we have

$$\int_{-1}^1 t^{2r}(1-t^2)^{\frac{d-3}{2}} \mathrm{d}t = \int_0^1 u^{r-\frac{1}{2}}(1-u)^{\frac{d-1}{2}-1} \mathrm{d}u = B\left(r + \frac{1}{2}, \frac{d-1}{2}\right) = \frac{\Gamma\left(r + \frac{1}{2}\right) \Gamma\left(\frac{d-1}{2}\right)}{\Gamma\left(r + \frac{d}{2}\right)}, \tag{15}$$

and

$$\frac{|\mathbb{S}^{d-2}|}{|\mathbb{S}^{d-1}|} = \frac{2\pi^{\frac{d-1}{2}}}{\Gamma\left(\frac{d-1}{2}\right)} \frac{\Gamma\left(\frac{d}{2}\right)}{2\pi^{\frac{d}{2}}} = \frac{\Gamma\left(\frac{d}{2}\right)}{\sqrt{\pi}\Gamma\left(\frac{d-1}{2}\right)}.$$

Combining both expressions yields

$$\mathbb{E}_{\boldsymbol{\xi} \sim \mathcal{U}_{\mathbb{S}^{d-1}}}\left[|\langle \boldsymbol{\xi}, x \rangle|^{2r}\right] = \|x\|^{2r} \frac{\Gamma(r + \frac{1}{2})}{\Gamma(r + \frac{d}{2})} \frac{\Gamma(\frac{d}{2})}{\sqrt{\pi}}, \tag{16}$$

and hence

$$\frac{\pi\Gamma\left(\frac{d+r}{2}\right)^2}{\Gamma\left(\frac{d}{2}\right)^2 \Gamma\left(\frac{r+1}{2}\right)^2} \mathbb{E}_{\boldsymbol{\xi} \sim \mathcal{U}_{\mathbb{S}^{d-1}}}\left[|\langle \boldsymbol{\xi}, x \rangle|^{2r}\right] = \frac{\|x\|^{2r}\sqrt{\pi}\Gamma(r + \frac{1}{2})}{\Gamma(\frac{r+1}{2})^2} \frac{\Gamma(\frac{d+r}{2})^2}{\Gamma(\frac{d}{2})\Gamma(r + \frac{d}{2})}.$$

The asymptotic expansion $\Gamma(z+a)/\Gamma(z) \sim z^a$ for $z \to \infty$, see (NIST, 5.11.12), implies that

$$\lim_{d \to \infty} \frac{\Gamma(\frac{d+r}{2})^2}{\Gamma(\frac{d}{2})\Gamma(r+\frac{d}{2})} = 1.$$

On the other hand, $\frac{\Gamma(\frac{d+r}{2})^2}{\Gamma(\frac{d}{2})\Gamma(r+\frac{d}{2})}$ is increasing with respect to $d$ because the identity $\Gamma(z+1) = z\Gamma(z)$ yields that

$$\frac{\Gamma(\frac{d+2+r}{2})^2}{\Gamma(\frac{d+2}{2})\Gamma(r+\frac{d+2}{2})} \bigg/ \frac{\Gamma(\frac{d+r}{2})^2}{\Gamma(\frac{d}{2})\Gamma(r+\frac{d}{2})} = \frac{\left(\frac{d+r}{2}\right)^2}{\left(\frac{d}{2}\right)\left(r+\frac{d}{2}\right)} = \frac{(d+r)^2}{2dr+d^2} \geq 1.$$

Hence, we see that

$$\frac{\Gamma(\frac{d+r}{2})^2}{\Gamma(\frac{d}{2})\Gamma(r+\frac{d}{2})} \leq 1,$$

and thus finally

$$\mathbb{V}_d[f] \leq \mathbb{E}_{\boldsymbol{\xi} \sim \mathcal{U}_{\mathbb{S}^{d-1}}}\left[f(|\langle \xi, x \rangle|)^2\right] = \frac{\pi\Gamma\left(\frac{d+r}{2}\right)^2}{\Gamma\left(\frac{d}{2}\right)^2\Gamma\left(\frac{r+1}{2}\right)^2}\mathbb{E}_{\boldsymbol{\xi} \sim \mathcal{U}_{\mathbb{S}^{d-1}}}\left[|\langle \boldsymbol{\xi}, x \rangle|^{2r}\right]$$

$$\leq \frac{\sqrt{\pi}\,\Gamma(r+\frac{1}{2})}{\Gamma(\frac{r+1}{2})^2}\|x\|^{2r},$$

which proves (7).

**iii):** We move on to the thin plate spline kernel with $f(t) = d|t|^2 \log(|t|) + C_d|t|^2$. As above, w.l.o.g. we assume that $x = se_d$ with $s \geq 0$. Then

$$\mathbb{E}\left[f(|\langle \xi, x \rangle|)^2\right]$$

$$= \frac{|\mathbb{S}^{d-2}|}{|\mathbb{S}^{d-1}|} 2 \int_0^1 \left(ds^2\xi_d^2\log(s\xi_d) - C_d s^2\xi_d^2\right)^2 (1-\xi_d^2)^{\frac{d-3}{2}} \mathrm{d}\xi_d$$

$$= \frac{\Gamma\left(\frac{d}{2}\right)}{\sqrt{\pi}\Gamma\left(\frac{d-1}{2}\right)} \frac{d^2}{2} \int_0^1 s^4\xi_d^4 \left(2\log\left(\xi_d\right) + 2\log(s) + H_{\frac{d}{2}} - 2 + \log(4)\right)^2 (1-\xi_d^2)^{\frac{d-3}{2}} \mathrm{d}\xi_d$$

$$= \frac{d^2\Gamma\left(\frac{d}{2}\right)}{2\sqrt{\pi}\Gamma\left(\frac{d-1}{2}\right)} s^4 \left(4\int_0^1 \xi_d^4\log^2(\xi_d)(1-\xi_d^2)^{\frac{d-3}{2}}\mathrm{d}\xi_d \right. \tag{17}$$

$$+ 4(2\log(s) + H_{\frac{d}{2}} - 2 + \log(4))\int_0^1 \xi_d^4\log(\xi_d)(1-\xi_d^2)^{\frac{d-3}{2}}\mathrm{d}\xi_d$$

$$\left. + \left(2\log(s) + H_{\frac{d}{2}} - 2 + \log(4)\right)^2 \int_0^1 \xi_d^4(1-\xi_d^2)^{\frac{d-3}{2}}\mathrm{d}\xi_d\right).$$

We analyze the terms separately. Denote by $\psi^{(n)}$ the $n$-th derivative of the digamma function and by $\gamma \approx 0.57$ the Euler–Mascheroni constant. In the following, we use the asymptotics

$$H_x = \log(x) + \gamma + \mathcal{O}(x^{-1}), \quad \psi^{(0)}(x) = \log(x) + \mathcal{O}(x^{-1}), \quad \psi^{(1)}(x) = \mathcal{O}\left(x^{-1}\right).$$

By (Gradshteyn & Ryzhik, 2007, 4.261.21), we have

$$\int_0^1 \xi_d^4\log^2(\xi_d)(1-\xi_d^2)^{\frac{d-3}{2}}\mathrm{d}\xi_d$$

$$= \frac{3\sqrt{\pi}\,\Gamma\left(\frac{d-1}{2}\right)}{32\Gamma\left(\frac{d+4}{2}\right)}\left(\psi^{(1)}(\tfrac{5}{2}) - \psi^{(1)}(\tfrac{d+4}{2}) + \left(\psi^{(0)}(\tfrac{5}{2}) - \psi^{(0)}(\tfrac{d+4}{2})\right)^2\right),$$

$$= \frac{3\sqrt{\pi}\,\Gamma\left(\frac{d-1}{2}\right)}{32\Gamma\left(\frac{d+4}{2}\right)}\left(\log(d)^2 - 2\left(\psi^{(0)}\left(\tfrac{5}{2}\right) + \log(2) + \mathcal{O}(\tfrac{1}{d})\right)\log(d)\right.$$

$$\left. + \psi^{(1)}\left(\tfrac{5}{2}\right) + \left(\psi^{(0)}\left(\tfrac{5}{2}\right) + \log(2)\right)^2 + \mathcal{O}\left(d^{-1}\right)\right),$$

and

$$\int_0^1 \xi_d^4 \log(\xi_d)(1 - \xi_d^2)^{\frac{d-3}{2}} \, \mathrm{d}\xi_d = -\frac{\sqrt{\pi}\Gamma\left(\frac{d-1}{2}\right)}{16\Gamma\left(\frac{d+4}{2}\right)} \left(-8 + 3H_{1+\frac{d}{2}} + \log(64)\right)$$

$$= -\frac{\sqrt{\pi}\Gamma\left(\frac{d-1}{2}\right)}{16\Gamma\left(\frac{d+4}{2}\right)} \left(3\log(d) + 3\gamma - 8 + \log(8) + \mathcal{O}(d^{-1})\right).$$

Recall from (15) that

$$\int_0^1 \xi_d^4 (1 - \xi_d^2)^{\frac{d-3}{2}} \, \mathrm{d}\xi_d = \frac{3\sqrt{\pi}\Gamma\left(\frac{d-1}{2}\right)}{8\Gamma\left(\frac{d+4}{2}\right)}.$$

Furthermore, using that $\log(4) = 2\log(2)$, we obtain

$$2\log(s) + H_{\frac{d}{2}} - 2 + \log(4) = 2\log(s) + \log(d) + \gamma + \log(2) - 2 + \mathcal{O}(d^{-1}),$$

$$\left(2\log(s) + H_{\frac{d}{2}} - 2 + \log(4)\right)^2 = \left(2\log(s) + \log(d) + \gamma + \log(2) - 2 + \mathcal{O}(d^{-1})\right)^2$$

$$= 4\log(s)^2 + \log(d)^2 + 4\log(s)\log(d)$$
$$+ 2(\gamma + \log(2) - 2)\log(d) + 4\log(s)(\gamma + \log(2) - 2)$$
$$+ (\gamma + \log(2) - 2)^2 + \mathcal{O}(d^{-1}).$$

Plugging this into (17) yields

$$4\frac{\Gamma\left(\frac{d+4}{2}\right)}{\sqrt{\pi}\Gamma\left(\frac{d-1}{2}\right)} \frac{\sqrt{\pi}\Gamma\left(\frac{d-1}{2}\right)}{\Gamma\left(\frac{d}{2}\right) d^2 s^4} \mathbb{E}\left[f(|\langle\xi, x\rangle|)^2\right] = \frac{1 + 2/d}{s^4}\mathbb{E}\left[f(|\langle\xi, x\rangle|)^2\right]$$

$$= \frac{3}{4}\left(\log(d)^2 - 2\left(\psi^{(0)}(\tfrac{5}{2}) + \log(2) + \mathcal{O}(\tfrac{1}{d})\right)\log(d) + \psi^{(1)}(\tfrac{5}{2}) + \left(\psi^{(0)}(\tfrac{5}{2}) + \log(2)\right)^2 + \mathcal{O}(\tfrac{1}{d})\right)$$

$$- \frac{1}{8}(2\log(s) + \log(d) + \gamma + \log(2) - 2 + \mathcal{O}(\tfrac{1}{d}))\left(3\log(d) + 3\gamma - 8 + \log(8) + \mathcal{O}(\tfrac{1}{d})\right)$$

$$+ \frac{3}{4}\left(4\log(s)^2 + \log(d)^2 + 4\log(s)\log(d) + 2(\gamma + \log(2) - 2)\log(d)\right.$$
$$\left. + 4\log(s)(\gamma + \log(2) - 2) + (\gamma + \log(2) - 2)^2 + \mathcal{O}\left(d^{-1}\right)\right)$$

$$= \log(d)^2\left(\frac{3}{4} - \frac{3}{2} + \frac{3}{4}\right)$$

$$+ \log(d)\left(-\frac{3}{2}\left(\psi^{(0)}(\tfrac{5}{2}) + \log(2) + \mathcal{O}(\tfrac{1}{d})\right) - \frac{1}{2}(3\gamma - 8 + \log(8))\right.$$

$$\left. - \frac{3}{2}((2\log(s) + \gamma + \log(2) - 2) + \frac{3}{4}(4\log(s) + 2(\gamma + \log(2) - 2))\right)$$

$$+ 3\log(s)^2 - \log(s)(3\gamma - 8 + \log(8))$$

$$+ \frac{3}{4}\left(\psi^{(1)}(\tfrac{5}{2}) + \left(\psi^{(0)}(\tfrac{5}{2}) + \log(2)\right)^2\right) + 2(\gamma + \log(2) - 2)^2 + \mathcal{O}\left(d^{-1}\right)$$

$$= \log(d)\left(-\frac{3}{2}\psi^{(0)}(\tfrac{5}{2}) - \frac{3}{2}\gamma - 3\log(2) + +\mathcal{O}(\tfrac{1}{d})\right) + 3\log(s)^2 - \log(s)(3\gamma + \log(8) - 8)$$

$$+ \frac{3}{4}\left(\psi^{(1)}(\tfrac{5}{2}) + \left(\psi^{(0)}(\tfrac{5}{2}) + \log(2)\right)^2\right) + 2(\gamma + \log(2) - 2)^2 + \mathcal{O}(d^{-1}).$$

With the identity $\psi^{(0)}(\tfrac{5}{2}) = -2\log(2) - \gamma + \tfrac{8}{3}$ and

$$c_1 := -3\gamma - \log(8) + 8 \approx 4.189$$

$$c_2 := \frac{3}{4}(\psi^{(1)}(\tfrac{5}{2}) + (\psi^{(0)}(\tfrac{5}{2}) + \log(2))^2) + 2(\gamma + \log(2) - 2)^2 \approx 2.895, \tag{18}$$

we obtain

$$\frac{1 + 2/d}{s^4}\mathbb{E}\left[f(|\langle\xi, x\rangle|)^2\right] = 3\log(s)^2 + c_1\log(s) + c_2 + \mathcal{O}\left(d^{-1}\log d\right).$$

**iv):** By (Hertrich, 2024, Thm 3), the transformed function has the form

$$f(s) = \sum_{n=0}^{\infty} b_n x^n \quad \text{with} \quad b_n = \frac{\sqrt{\pi}\Gamma\left(\frac{n+d}{2}\right)}{\Gamma\left(\frac{d}{2}\right)\Gamma\left(\frac{n+1}{2}\right)} a_n.$$

Moreover,

$$f(s)^2 = \sum_{n=0}^{\infty} c_n s^n := \sum_{n=0}^{\infty} \left(\sum_{k=0}^{n} b_k b_{n-k}\right) s^n$$

and, by (16),

$$\mathbb{E}_{\xi \sim \mathcal{U}_{\mathbb{S}^{d-1}}}\left[f(|\langle \xi, x\rangle|)^2\right] = \sum_{n=0}^{\infty} \frac{\Gamma\left(\frac{d}{2}\right)\Gamma\left(\frac{n+1}{2}\right)}{\sqrt{\pi}\Gamma\left(\frac{n+d}{2}\right)} c_n \|x\|^n.$$

Using that $d$ is odd and applying the identities $\Gamma(z+1) = \Gamma(z)z$ and $\Gamma(\frac{1}{2}) = \sqrt{\pi}$, we have

$$\tilde{c}_{k,n,d} := \frac{\Gamma\left(\frac{d}{2}\right)\Gamma\left(\frac{n+1}{2}\right)}{\sqrt{\pi}\Gamma\left(\frac{n+d}{2}\right)} \frac{\sqrt{\pi}\Gamma\left(\frac{k+d}{2}\right)}{\Gamma\left(\frac{d}{2}\right)\Gamma\left(\frac{k+1}{2}\right)} \frac{\sqrt{\pi}\Gamma\left(\frac{n-k+d}{2}\right)}{\Gamma\left(\frac{d}{2}\right)\Gamma\left(\frac{n-k+1}{2}\right)}$$

$$= \frac{\Gamma\left(\frac{n+1}{2}\right)}{\Gamma\left(\frac{n+1+d-1}{2}\right)} \frac{\Gamma\left(\frac{k+1+d-1}{2}\right)}{\Gamma\left(\frac{k+1}{2}\right)} \frac{\sqrt{\pi}\Gamma\left(\frac{n-k+1+d-1}{2}\right)}{\Gamma\left(\frac{d}{2}\right)\Gamma\left(\frac{n-k+1}{2}\right)}$$

$$= \prod_{j=1}^{\frac{d-1}{2}} \frac{(k+2j-1)(n-k+2j-1)}{(n+2j-1)(2j-1)}.$$

Hence, we obtain

$$\mathbb{V}_d[f](x) = \sum_{n=0}^{\infty} \left(\sum_{k=0}^{n} (\tilde{c}_{k,n,d} - 1) a_k a_{n-k}\right) \|x\|^n.$$

We apply the identity $4xy = (x+y)^2 - (x-y)^2$ to the numerator (with $x = k+2j-1$, $y = n-k+2j-1$) and denominator (with $x = n+2j-1$, $y = 2j-1$), and obtain

$$\tilde{c}_{k,n,d} = \prod_{j=1}^{\frac{d-1}{2}} \frac{(k+2j-1)(n-k+2j-1)}{(n+2j-1)(2j-1)} = \prod_{j=1}^{\frac{d-1}{2}} \frac{(n+2(2j-1))^2 - (n-2k)^2}{(n+2(2j-1))^2 - n^2}$$

$$= \prod_{j=1}^{\frac{d-1}{2}} \left(1 + \frac{n^2 - (n-2k)^2}{(n+2(2j-1))^2 - n^2}\right) = \prod_{j=1}^{\frac{d-1}{2}} \left(1 + \frac{k(n-k)}{(2j-1)(n+2j-1)}\right).$$

For $F$ the Laplace kernel, assuming w.l.o.g. that $\alpha = 1$, we have $a_n := (-1)^n/n!$. Consequently,

$$\mathbb{V}_3[f](x) = \sum_{n=0}^{\infty} \sum_{k=0}^{n} \frac{k(n-k)}{n+1} \frac{1}{k!(n-k)!}(-\|x\|)^n = \sum_{n=0}^{\infty} \frac{(-\|x\|)^n}{(n+1)!} \sum_{k=0}^{n} \binom{n}{k}(kn - k^2)$$

$$= \sum_{n=0}^{\infty} \frac{(-\|x\|)^n}{(n+1)!}\left(n^2 2^{n-1} - n(n+1)2^{n-2}\right) = \frac{1}{4}\sum_{n=0}^{\infty} \frac{(-2\|x\|)^n}{(n+1)!}n(n-1)$$

$$= \frac{1}{4}\sum_{n=2}^{\infty} \frac{(-2\|x\|)^n}{(n+1)!}(n+1-2)n = \frac{1}{4}\sum_{n=2}^{\infty} \frac{(-2\|x\|)^n}{(n-1)!} - \frac{1}{2}\sum_{n=2}^{\infty} \frac{(-2\|x\|)^n}{(n+1)!}n. \quad (19)$$

For the first term, we obtain

$$\sum_{n=2}^{\infty} \frac{(-2\|x\|)^n}{(n-1)!} = -2\|x\|\sum_{n=1}^{\infty} \frac{(-2\|x\|)^n}{n!} = -2\|x\|\left(e^{-2\|x\|} - 1\right), \quad (20)$$

and for the second

$$\sum_{n=2}^{\infty} \frac{(-2\|x\|)^n}{(n+1)!}(n+1-1) = \sum_{n=2}^{\infty} \frac{(-2\|x\|)^n}{n!} - \sum_{n=2}^{\infty} \frac{(-2\|x\|)^n}{(n+1)!}$$

$$= e^{-2\|x\|} - 1 + 2\|x\| + \frac{1}{2\|x\|}\left(e^{-2\|x\|} - 1 + 2\|x\| - 2\|x\|^2\right). \quad (21)$$

Plugging (20) and (21) into (19) finally yields

$$
\begin{aligned}
\mathbb{V}_3[f](x) &= \frac{-\|x\|}{2}(\mathrm{e}^{-2\|x\|} - 1) - \frac{1}{2}\left(\mathrm{e}^{-2\|x\|} - 1 + 2\|x\| + \frac{\mathrm{e}^{-2\|x\|}}{2\|x\|} - \frac{1}{2\|x\|} + 1 - \|x\|\right) \\
&= \frac{\mathrm{e}^{-2\|x\|}}{4\|x\|}\left(\mathrm{e}^{2\|x\|} - (2\|x\|^2 + 2\|x\| + 1)\right) \\
&= \frac{1}{4\|x\|}\left(\mathrm{e}^{2\|x\|} - (2\|x\|^2 + 2\|x\| + 1)\right)F(\|x\|)^2.
\end{aligned}
$$

For the Gauss kernel, we can follow exactly the same lines, after considering w.l.o.g. $\sigma^2 = 1/2$ and replacing $\|x\|$ with $\|x\|^2$.

## D  DISCUSSION OF THEOREM 1

### D.1  COMPARISON TO THE ERROR BOUNDS FROM HERTRICH (2024)

In the following, we give a short summary how Theorem 1 improves the error bounds from Hertrich (2024).

- For positive definite kernels and Riesz kernels, the bound from Theorem 1 is dimension-independent. In contrast, Hertrich (2024) only proves a dimension-independent absolute error bound for the Gauss kernel (and conjectures that this is also true for the Laplace and Matérn kernel). For the Riesz kernel, the error bound in Hertrich (2024) depends on the dimension by $\mathcal{O}(\sqrt{d})$.

- Theorem 1 bounds the error of the thin plate spline kernel. For this kernel Hertrich (2024) does not provide a bound.

- For kernels with analytic basis functions and for the Riesz kernel, we exactly compute the variance $\mathbb{V}_d[f]$. In particular, Theorem 1 provides an *exact* calculation for the mean square error, i.e., no tighter estimation is possible. E.g., for the Gauss and Laplace kernel, this yields an improvement of the absolute error bounds given in Hertrich (2024), as we observe that this bounds actually decay to zero for $\|x\| \to \infty$.

### D.2  THE VARIANCE OF GAUSS AND LAPLACE KERNEL FOR $d > 3$

For $d > 3$, $d$ odd, we evaluated $\mathbb{V}_d(f)$ for $d = 5, ..., 15$, and make the following conjecture. Let $T_n(f) = \sum_{k=0}^{n} \frac{f^{(k)}(0)}{k!}x^k$ denote the Taylor expansion of $f$ in zero up to order $n$. We conjecture that, for $d \geq 3$, $d$ odd, and the Laplace kernel $F(x) = \mathrm{e}^{-\alpha x}$, it holds that

$$
\mathbb{V}_d[f](x) = \frac{\mathrm{e}^{-2\alpha\|x\|}}{c_d(\alpha\|x\|)^{d-2}}\left(\sum_{i=0}^{\frac{d-3}{2}}(\alpha\|x\|)^{2i}(-1)^{\frac{d-5}{2}-i}c_{i,d}T_{d-2i}\left(\mathrm{e}^{2\alpha\|x\|}\right) + \sum_{i=0}^{\frac{d-5}{2}}b_{i,d}(-1)^i(\alpha\|x\|)^{d+i}\right),
$$

where

$$
\begin{aligned}
c_d &= (12 + 4(d-5))\,c_{d-2} \quad \text{for} \quad d \geq 5 \quad \text{with} \quad c_3 = 4, \\
c_{0,d} &= (d-2)^2(d-4)c_{0,d-2} \quad \text{for} \quad d \geq 5 \quad \text{with} \quad c_{0,3} = 1, \\
b_{\frac{d-5}{2},d} &= 2b_{\frac{d-5}{2},d-2} \quad \text{for} \quad d \geq 7 \quad \text{with} \quad b_{0,5} = 4.
\end{aligned}
$$

For the remaining coefficients, we have not found a general simple rule. Our numerical computations indicate that they are positive and that $\mathbb{V}_d[f](x)\|x\|$ is monotonically increasing in $\|x\|$, with upper bound $s_d/\alpha$ and quadratic convergence to 0 for $x \to 0$. The upper bound $s_d$ increases slowly in $d$, and our simulations hint that it might be bounded by a constant independently of $d$, see Figure 3. For the Gauss kernel, the variance has the same form, except that $\alpha^{-1}$ is replaced with $\sigma/\sqrt{2}$ and $\|x\|$ with $\|x\|^2$ at all occurrences.

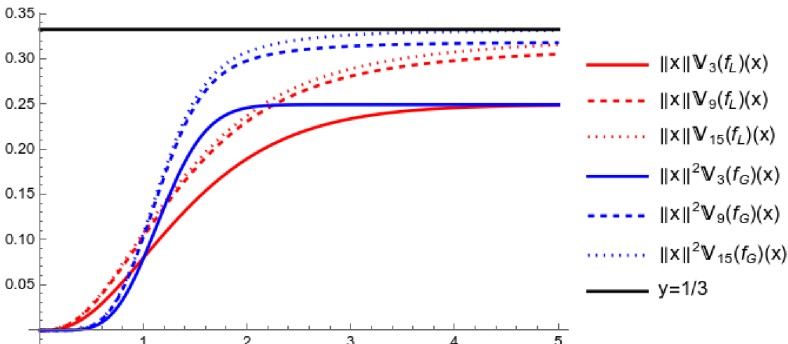

Figure 3: The scaled variance of the Laplace kernel $f_L$ with $\alpha = 1$ (in red) and the Gauss kernel $f_G$ with $\sigma = 1/\sqrt{2}$ (in blue) for dimension $d = 3$ (solid lines), $d = 9$ (dashed lines) and $d = 15$ (dotted lines). The variance is multiplied times $\|x\|$ (Laplace kernel) and $\|x\|^2$ (Gauss kernel). We observe that the scaled variance increases monotonically in all cases, seemingly bounded from above by a constant (black solid line).

## E  RANDOMIZATION OF A QMC SEQUENCE

Having a QMC sequence $(\boldsymbol{\xi}^P)_P$, see Section 3.1, we can easily obtain an unbiased estimator in (4) by considering the QMC sequence $(A\boldsymbol{\xi}^P)_P$ with a uniformly chosen orthogonal matrix $A \sim \mathcal{U}_{\mathrm{O}(d)}$ because, according to (Ragozin, 1974, (2.3)), we have

$$\mathbb{E}_{A \sim \mathcal{U}_{\mathrm{O}(d)}}[\psi(A\xi_p^P)] = \int_{\mathrm{O}(d)} \psi(A\xi_p^P)\mathrm{d}\mathcal{U}_{\mathrm{O}(d)}(A) = \int_{\mathbb{S}^{d-1}} \psi(\eta)\mathrm{d}\mathcal{U}_{\mathbb{S}^{d-1}}(\eta),$$

where $\mathcal{U}_{\mathrm{O}(d)}$ is the uniform distribution on the set $\mathrm{O}(d)$ of orthogonal $d \times d$ matrices. Furthermore, note that $\frac{1}{P}\sum_{i=1}^{P} f(\xi_i^P)$ converges to $\int_{\mathbb{S}^{d-1}} f(\xi)\mathrm{d}\mathcal{U}_{\mathbb{S}^{d-1}}(\xi)$ for $P \to \infty$ and all continuous functions $f$ because $H^s(\mathbb{S}^{d-1})$ is dense in $C(\mathbb{S}^{d-1})$ and the weights are all one, cf. (Atkinson, 1991, Sect 5.2).

## F  PROOF OF THEOREM 3

**Gauss kernel:** A sufficient criterion for the function $g_x$ being in the Sobolev space $H^s(\mathbb{S}^{d-1})$ is that $f(|\cdot|)$ is $s$ times continuously differentiable. For $F(t) = \exp(-\frac{t^2}{2\sigma^2})$, the one-dimensional basis function is given by the confluent hypergeometric function $f(t) = {}_1F_1(\frac{d}{2};\frac{1}{2};\frac{-t^2}{2\sigma^2})$. We know by (Hertrich, 2024, Lem 6) that $f = \mathcal{F}_1(g)$ with $g(\omega) = \frac{d\pi\sigma\exp(-2\pi^2\sigma^2\omega^2)(2\pi^2\sigma^2\omega^2)^{(d-1)/2}}{\sqrt{2}\Gamma(\frac{d+2}{2})}$, where $\mathcal{F}_1$ denotes the one-dimensional Fourier transform. Then, $g_k(\omega) := \omega^k g(\omega)$ is absolutely integrable for any $k \in \mathbb{N}$ such that the differentiation/multiplication formula for the Fourier transform yields that the $k$-th derivative of $f$ exists and is given by the continuous function $f^{(k)} = (2\pi\mathrm{i})^k\mathcal{F}_1(g_k)$.

**Riesz kernel:** With fixed $x \in \mathbb{R}^d \setminus \{0\}$, we have $g_x(\xi) = \frac{\sqrt{\pi}\Gamma(\frac{d+r}{2})}{\Gamma(\frac{d}{2})\Gamma(\frac{r+1}{2})}|\langle\xi,x\rangle|^r$. We set $\eta := \frac{x}{\|x\|} \in \mathbb{S}^{d-1}$, then $x = \|x\|\eta$ and accordingly $g_x(\xi) = \|x\|^r g_\eta(\xi)$. The Sobolev norm (14) reads

$$\|g_x\|^2_{H^s(\mathbb{S}^{d-1})} = \sum_{n=0}^{\infty}\sum_{k=1}^{N_{n,d}}\left(n+\frac{d-2}{2}\right)^{2s}\left|\|x\|^r\int_{\mathbb{S}^{d-1}}g_\eta(\xi)\overline{Y_n^k(\xi)}\mathrm{d}\xi\right|^2.$$

The last integral can be computed with the help of the so-called $\lambda$-cosine transform. By (Rubin, 2014, (3.5) and (3.8)), the $\lambda$-cosine transform of a function $h \in L^1(\mathbb{S}^{d-1})$ for $\lambda > -1$ is defined by

$$\mathcal{C}^\lambda[h](\omega) = \frac{1}{|\mathbb{S}^{d-1}|}\frac{\sqrt{\pi}\Gamma(-\frac{\lambda}{2})}{\Gamma(\frac{d}{2})\Gamma(\frac{\lambda+1}{2})}\int_{\mathbb{S}^{d-1}}h(\theta)|\langle\omega,\theta\rangle|^\lambda\mathrm{d}\theta,$$

for $\omega \in \mathbb{S}^{d-1}$ and satisfies

$$\mathcal{C}^{\lambda}[Y_n^k](\omega) = Y_n^k(\omega) \begin{cases} (-1)^{\frac{n}{2}} \frac{\Gamma(\frac{n-\lambda}{2})}{\Gamma(\frac{n+d+\lambda}{2})}, & n \text{ even}, \\ 0, & n \text{ odd}. \end{cases}$$

Hence, we obtain that

$$\int_{\mathbb{S}^{d-1}} g_\eta(\xi)\overline{Y_n^k(\xi)}\mathrm{d}\xi = \frac{|\mathbb{S}^{d-1}|\Gamma(\frac{d+r}{2})}{\Gamma(-\frac{r}{2})}\mathcal{C}^r\left[\overline{Y_n^k}\right](\eta) = (-1)^{\frac{n}{2}}\frac{|\mathbb{S}^{d-1}|\Gamma(\frac{n-r}{2})\Gamma(\frac{d+r}{2})}{\Gamma(\frac{n+d+r}{2})\Gamma(-\frac{r}{2})}\overline{Y_n^k}(\eta)$$

if $n$ is even and 0 if $n$ is odd. The addition formula for spherical harmonics,

$$\sum_{k=1}^{N_{n,d}} |Y_n^k(\eta)|^2 = \frac{N_{n,d}}{|\mathbb{S}^{d-1}|},$$

see Atkinson & Han (2012), and the substitution $n = 2m$ yield

$$\|g_x\|_{H^s(\mathbb{S}^{d-1})}^2 = \sum_{m=0}^{\infty} \left(2m + \frac{d-2}{2}\right)^{2s} \|x\|^{2r} \left(\frac{\Gamma(m-\frac{r}{2})\Gamma(\frac{d+r}{2})}{\Gamma(m+\frac{d+r}{2})\Gamma(-\frac{r}{2})}\right)^2 |\mathbb{S}^{d-1}|N_{2m,d}.$$

The asymptotic relations $N_{n,d} \sim n^{d-2}$ from (13) and $\Gamma(z+a)/\Gamma(z) \sim z^a$ for $z \to \infty$, see (NIST, 5.11.12), yield that the $m$-th summand behaves asymptotically like a multiple of $(2m)^{2s-2r-2}$, therefore the series converges if and only if $s < r + \frac{1}{2}$.

**Matérn kernel:** The one-dimensional basis function of the Matérn kernel is (Hertrich, 2024, Appx C.1)

$$f(t) = \underbrace{{}_1F_2\left(\frac{d}{2};\frac{1}{2},1-\nu;\frac{\nu t^2}{2\beta^2}\right)}_{=:f_1(t)} - \underbrace{|t|^{2\nu}}_{=:f_2(t)} \underbrace{\frac{\Gamma(1-\nu)\Gamma(\nu+\frac{d}{2}(2\nu)^\nu)}{\Gamma(\frac{d}{2})\Gamma(2\nu+1)\beta^{2\nu}} {}_1F_2\left(\nu+\frac{d}{2};\nu+\frac{1}{2},\nu+1;\frac{\nu t^2}{2\beta^2}\right)}_{=:f_3(t)}.$$

The above hypergeometric functions $f_1$ and $f_3$, whose parameters do not contain any nonpositive integers, are entire functions in $\mathbb{R}$. Hence, the corresponding spherical functions $(g_1)_x$ and $(g_3)_x$ from (10) are in $C^\infty(\mathbb{S}^{d-1})$, because they can be extended to smooth functions defined on a neighborhood of $\mathbb{S}^{d-1}$. Furthermore, since $f_2$ is a multiple of the one-dimensional basis function of the Riesz kernel, we know from above that $(g_2)_x \in H^s(\mathbb{S}^{d-1})$ for $s < 2\nu + \frac{1}{2}$. Hence, by (Quellmalz, 2020, Thm 5.2), we see that the product $(g_2)_x(g_3)_x$ is also in $H^s(\mathbb{S}^{d-1})$. Note that the referenced theorem is only formulated for in integer $s$, but this can be easily extended using an interpolation argument (Quellmalz, 2020, Sect 5.3), because the multiplication with a smooth function constitutes a continuous operator both $H^{\lfloor s \rfloor}(\mathbb{S}^{d-1}) \to H^{\lfloor s \rfloor}(\mathbb{S}^{d-1})$ and $H^{\lfloor s \rfloor+1}(\mathbb{S}^{d-1}) \to H^{\lfloor s \rfloor+1}(\mathbb{S}^{d-1})$.

## G   BACKGROUND ON RFF AND RELATION WITH SLICING

We denote by $\mathcal{M}_+(\mathbb{R}^d)$ the space of finite positive Borel measures on $\mathbb{R}^d$. Such measures can be identified with linear functional on the space $C_0(\mathbb{R}^d)$ of continuous functions that vanish at infinity. The Fourier transform of measures is a linear operator defined by

$$\mathcal{F}_d \colon \mathcal{M}_+(\mathbb{R}^d) \to C_0(\mathbb{R}^d), \quad \mathcal{F}_d[\mu](x) = \int_{\mathbb{R}^d} \mathrm{e}^{-2\pi\mathrm{i}\langle x,v\rangle}\mathrm{d}\mu(v),$$

cf. (Plonka et al., 2023, Sect 4.4). By Bochner's theorem, the Fourier transform is bijective from $\mathcal{M}_+(\mathbb{R}^d)$ to the set of positive definite functions on $\mathbb{R}^d$, see ii) in Section 2 for the definition. If $\mu$ is a probability measure, i.e., $\mu(\mathbb{R}^d) = 1$, we have $\mathcal{F}_d[\mu](0) = 1$.

In the following, let $F \circ \|\cdot\|$ be a positive definite function on $\mathbb{R}^d$ with $F(0) = 1$.

**RFF:** Random Fourier features (RFF), see Rahimi & Recht (2007), use that, by Bochner's theorem, $F(\|\cdot\|)$ is the Fourier transform of a probability measure $\mu \in \mathcal{M}_+(\mathbb{R}^d)$, i.e.,

$$F(\|x\|) = \mathcal{F}_d[\mu](x) = \mathbb{E}_{v\sim\mu}\left[\exp(2\pi\mathrm{i}\langle x,v\rangle)\right] \approx \frac{1}{D}\sum_{p=1}^{D}\exp(2\pi\mathrm{i}\langle x,v_p\rangle), \tag{22}$$

where we sample $D \in \mathbb{N}$ instances $v_p$ iid from $\mu$. Using that $F$ is real-valued, we can replace the exponential by the cosine in (22). By (Altekrüger et al., 2023, Prop C.2), the radial measure $\mu$ can be decomposed as follows. We define $\iota \colon \mathbb{R}^d \to [0, \infty)$, $\iota(x) = \|x\|$ and its pushforward measure $\tilde{\mu} \coloneqq \iota_* \mu = \mu \circ \iota^{-1} \in \mathcal{M}_+([0, \infty))$, then we have

$$\mu = T(\tilde{\mu} \otimes \mathcal{U}_{\mathbb{S}^{d-1}}), \quad \text{where} \quad T(r, \xi) = r\xi.$$

Hence, sampling from $\mu$ can be realized by $v_p = r_p \xi_p$, where $\xi_p \sim \mathcal{U}_{\mathbb{S}^{d-1}}$ and $r_p \sim \tilde{\mu}$. Then (22) becomes

$$F(\|x\|) \approx \frac{1}{D} \sum_{p=1}^{D} \cos(2\pi r_p \langle x, \xi_p \rangle). \tag{23}$$

**RFF Summation:** For approximating the kernel sum (1), we insert the RFF (22) and obtain

$$s_m \approx \sum_{n=1}^{N} w_n \frac{1}{D} \sum_{p=1}^{D} e^{2\pi i \langle y_m - x_n, v_p \rangle} = \frac{1}{D} \sum_{p=1}^{D} e^{2\pi i \langle y_m, v_p \rangle} \sum_{n=1}^{N} w_n e^{-2\pi i \langle x_n, v_p \rangle}, \tag{24}$$

where $v_1, \ldots, v_D$ are iid samples of $\mu$. Since the inner sum over $n$ is independent of $m$ and therefore has to be evaluated only $D$ times, the total computational complexity of computing (24) for all $m = 1, ..., M$ is $\mathcal{O}(D(N + M))$.

**Slicing:** The slicing approach uses the approximation (4), i.e.,

$$F(\|x\|) \approx \frac{1}{P} \sum_{p=1}^{P} f(|\langle x, \xi_p \rangle|),$$

where $\xi_p \sim \mathcal{U}_{\mathbb{S}^{d-1}}$. By (Rux et al., 2025, Cor 4.11), the function $f(|\cdot|)$ is positive definite and hence possesses a Fourier transform, which is a probability measure on $\mathbb{R}$ because $f(0) = F(0) = 1$. Applying RFF with $Q$ points to the one-dimensional function $f(|\cdot|)$, we obtain

$$F(\|x\|) \approx \frac{1}{PQ} \sum_{p=1}^{P} \sum_{q=1}^{Q} \cos(2\pi r_{p,q} \langle x, \xi_p \rangle), \tag{25}$$

where $r_{p,q} \sim \mathcal{F}_1^{-1}[f(|\cdot|)]$. According to (Rux et al., 2025, Cor 4.11), we have

$$\mathcal{F}_1^{-1}[f(|\cdot|)] = \mathcal{A}_1^* \iota_* \mu = \mathcal{A}_1^* \tilde{\mu},$$

where $\mathcal{A}_1^* \colon \mathcal{M}_+([0, \infty)) \to \mathcal{M}_+(\mathbb{R})$ is the symmetrization operator that extends a measure on $[0, \infty)$ to an even measure on the real line and is defined for any $\nu \in \mathcal{M}_+([0, \infty))$ and $g \in C_0(\mathbb{R})$ by $\langle \mathcal{A}_1^* \nu, g \rangle = \langle \nu, g + g(-\cdot) \rangle$. Since the right-hand side of (25) is independent of the sign of $r_{p,q}$, it stays the same when we sample $r_{p,q}$ from $\tilde{\mu}$ instead of $\mathcal{A}_1^* \tilde{\mu}$. Therefore, if $Q = 1$, we see that the right-hand side of (25) is the same as the right-hand side of (23). In particular, RFF can be viewed as a special case of slicing. The respective slicing summation is given by (5) combined with the one-dimensional summation in Appendix I below.

## H  RELATION BETWEEN DISTANCE QMC DESIGNS AND ORTHOGONAL POINTS

Consider the QMC design $\boldsymbol{\xi}^P$ that is a minimizer (9) for $s = \frac{d}{2}$. To improve the error, Womersley (2018) suggested symmetric QMC designs, meaning that for every point $\xi$ it contains also the antipodal point $-\xi$. However, since the function (10) we want to integrate is symmetric, i.e., $g_x(\xi) = g_x(-\xi)$, we can discard one of the antipodal points $\xi$ and $-\xi$ and get the same result. Therefore, we minimize the functional

$$\mathcal{E}_{\text{sym}}(\boldsymbol{\xi}^P) = \mathcal{E}((\boldsymbol{\xi}^P, -\boldsymbol{\xi}^P)) = -2 \sum_{p,q=1}^{P} \left( \|\xi_p^P - \xi_q^P\| + \|\xi_p^P + \xi_q^P\| \right).$$

Our numerical trials indicate that indeed the minimizers of $\mathcal{E}_{\mathrm{sym}}$ yield a smaller integration error than the minimizers of $\mathcal{E}$. Using $\|\xi_p\| = 1$, we see that

$$-\left(\|\xi_p^P - \xi_q^P\| + \|\xi_p^P + \xi_q^P\|\right)^2 = -\left(\sqrt{2 - 2\langle\xi_p^P, \xi_q^P\rangle} + \sqrt{2 + 2\langle\xi_p^P, \xi_q^P\rangle}\right)^2$$
$$= -4 - 2\sqrt{4 - 4\langle\xi_p^P, \xi_q^P\rangle^2},$$

attains its minimum if and only if $\langle\xi_p^P, \xi_q^P\rangle = 0$. Hence, if $P \leq d$ and $\boldsymbol{\xi}^P$ is an orthonormal system in $\mathbb{R}^d$, then $\boldsymbol{\xi}^P$ a minimizer of $\mathcal{E}_{\mathrm{sym}}$. However, this argumentation does not work if $P > d$, as we can only choose $d$ orthogonal vectors.

# I  BACKGROUND ON ONE-DIMENSIONAL FAST FOURIER SUMMATION

In this section, we review literature about one-dimensional fast Fourier summation used in Section 4.3 and specify the parameters used in our numerical examples. Fast Fourier summations were proposed in Kunis et al. (2006); Potts et al. (2004) based on the non-equispaced fast Fourier transform (Beylkin, 1995; Dutt & Rokhlin, 1993), which is implemented in several libraries (Knopp et al., 2023; Keiner et al., 2009; Shih et al., 2021). In our numerical examples, we use the Julia library Knopp et al. (2023). Here, we follow a similar workflow as in Hertrich (2024).

Let $x_1, ..., x_N \in \mathbb{R}^d$, $y_1, ..., y_M \in \mathbb{R}^d$, $w_1, ..., w_N \in \mathbb{R}$ and $\mathrm{k}(x, y) = f(|x - y|) = g(x - y)$. We want to compute for $\xi \in \mathbb{S}^{d-1}$, $x_{n,\xi} := \langle x_n, \xi\rangle$ and $y_{m,\xi} := \langle y_m, \xi\rangle$ the one-dimensional kernel sums

$$t_m = \sum_{n=1}^N w_n \mathrm{k}(x_{n,\xi}, y_{m,\xi}) = \sum_{n=1}^N w_n g(x_{n,\xi} - y_{m,\xi}).$$

**Step 1: Rescaling**  For Step 2 and 3, we will need two properties. First, since we will use discrete (fast) Fourier transforms, we require that $x_{n,\xi}, y_{m,\xi}, x_{n,\xi} - y_{m,\xi} \in [-\frac{1}{2}, \frac{1}{2})$. For the important example of positive definite kernels, which decay to zero, we often can derive explicit formulas for the Fourier transform of $g$ via Bochner's integral and Rux et al. (2025), see Table 2 for some examples. In order to use this explicit formula for the fast Fourier summation, we will additionally require that $g(x) \approx 0$ for $|x| > \frac{1}{2}$.

Both properties can be achieved by rescaling the problem. More precisely, let $T < 0.5$ and $c = \max_{n=1...,N} \|x_n\| + \max_{m=1,...,M} \|y_m\|$. For the case of decaying positive definite kernels, assume that $g(x) \approx 0$ for $|x| > g_{\max}$. Then, it holds that

$$t_m = \sum_{n=1}^N w_n g(x_{n,\xi} - y_{m,\xi}) = \sum_{n=1}^N w_n \tilde{g}(\tilde{x}_{n,\xi} - \tilde{y}_{m,\xi}),$$

with

$$\tilde{g}(x) := g\left(\frac{x}{\tau}\right), \quad \tilde{x}_{n,\xi} := \tau x_{n,\xi} = \langle\tau x_n, \xi\rangle, \quad \tilde{y}_{n,\xi} := \tau y_{n,\xi} = \langle\tau y_n, \xi\rangle,$$

where the constant $\tau := \min\left\{\frac{T}{c}, \frac{1}{2g_{\max}}\right\}$ does not depend on $\xi$. Then, by definition, the two properties from above are fulfilled. For the rest of the section, we will denote the rescaled points $\tilde{x}_{n,\xi}, \tilde{y}_{m,\xi}$ and the rescaled kernel function $\tilde{g}$ again by $x_{n,\xi}, y_{m,\xi}$ and $g$.

In the case of the Gauss, Laplace or Matérn kernels, the rescaled kernel $\tilde{\mathrm{k}}(x, y) = \tilde{g}(x - y)$ is again a Gauss, Laplace or Matérn kernel with the altered parameter $\tilde{\sigma} = \tau\sigma$, $\tilde{\alpha} = \alpha/\tau$ or $\tilde{\beta} = \tau\beta$. In our numerics, we set $g_{\max} = 5m$ with $m = \sigma = \beta = \frac{1}{\alpha}$ for the Gauss, Laplace and Matérn kernel. Moreover, we set the threshold $T$ to $0.3$ for the Gauss kernel, to $0.2$ for the Matérn kernel and to $0.1$ for the Laplace kernel.

**Step 2: Computation of the Fourier Coefficients of the Kernel**  In the next step, we expand $g$ into its Fourier series on $[-\frac{1}{2}, \frac{1}{2})$ and truncate it by

$$g(x) = \sum_{k \in \mathbb{Z}} c_k(g) e^{2\pi i k x} \approx \sum_{k=-N_{\mathrm{ft}}/2}^{N_{\mathrm{ft}}/2-1} c_k(g) e^{2\pi i k x}, \quad c_k(g) = \int_{-\frac{1}{2}}^{\frac{1}{2}} g(x) e^{-2\pi i k x} \mathrm{d}x$$

with some even $N_{\text{ft}} \in \mathbb{N}$. To compute the Fourier coefficients $c_k(g)$, we employ that $g(x) \approx 0$ for $|x| > \frac{1}{2}$ such that Poisson's summation formula (see, e.g., Plonka et al., 2023, Thm 2.28) implies

$$c_k(g) \approx c_k \left( \sum_{l \in \mathbb{Z}} g(\cdot + l) \right) = \mathcal{F}_1[g](k),$$

where $\mathcal{F}_1[g](\omega)$ is the Fourier transform (11).

In our experiments, we choose $N_{\text{ft}} = 128$ for the Gauss, $N_{\text{ft}} = 512$ for the Matérn and $N_{\text{ft}} = 1024$ for the Laplace kernel. Note that the coefficients $c_k(g)$ do not depend on the input points and need to be computed only once for different choices of $\xi$. The function $\mathcal{F}_1[g]$ is analytically given for the Gauss, Laplace and Matérn kernel in Table 2.

**Step 3: Fast Fourier Summation** Finally, we use this expansion to compute the kernel sums

$$t_m \approx \sum_{n=1}^{N} \sum_{k=-N_{\text{ft}}/2}^{N_{\text{ft}}/2-1} w_n c_k(g) \mathrm{e}^{2\pi \mathrm{i} k(y_{m,\xi}-x_{n,\xi})} = \sum_{k=-N_{\text{ft}}/2}^{N_{\text{ft}}/2-1} c_k(g) \mathrm{e}^{2\pi \mathrm{i} k y_{m,\xi}} \underbrace{\sum_{n=1}^{N} w_n \mathrm{e}^{-2\pi \mathrm{i} k x_{n,\xi}}}_{=:\hat{w}_k} . \quad (26)$$

The computation of the second sum $\hat{w}_k$ is the adjoint discrete Fourier transform of the vector $w = (w_1, ..., w_N)$ at the non-equispaced knots $(-x_{1,\xi}, ..., -x_{N,\xi})$. Afterward, the computation of the vector $t = (t_1, ..., t_M)$ is the Fourier transform of the vector $(c_k(g) \hat{w}_k)_{k=-N_{\text{ft}}/2}^{N_{\text{ft}}/2-1}$ at the non-equispaced knots $(-y_{1,\xi}, ..., -y_{M,\xi})$. These Fourier transforms can be computed by the NFFT in time complexity $\mathcal{O}(N + N_{\text{ft}} \log N_{\text{ft}})$ and $\mathcal{O}(M + N_{\text{ft}} \log N_{\text{ft}})$ leading to an overall complexity of $\mathcal{O}(N + M + N_{\text{ft}} \log N_{\text{ft}})$.

## J COMPUTATIONAL COMPLEXITY

We consider the computational complexity of the random Fourier features (RFF) and the Fourier slicing summation. We denote the RFF summation (24) with $D$ features by

$$\tilde{s}_m^{\text{RFF}} := \sum_{n=1}^{N} w_n \frac{1}{D} \sum_{p=1}^{D} \mathrm{e}^{2\pi \mathrm{i} \langle y_m - x_n, v_p \rangle} = \frac{1}{D} \sum_{p=1}^{D} \mathrm{e}^{2\pi \mathrm{i} \langle y_m, v_p \rangle} \sum_{n=1}^{N} w_n \mathrm{e}^{-2\pi \mathrm{i} \langle x_n, v_p \rangle}. \quad (27)$$

Computing $\tilde{s}_m^{\text{RFF}}$ for all $m = 1, \dots, M$ has a complexity of $\mathcal{O}(D(N + M))$.

We consider the Fourier slicing as described in Appendix I, with the slight modification that instead of rescaling the problem, we take a $2T$-periodic Fourier series. To this end, we choose

$$R \geq \max_{n=1,...,N} \|x_n\| + \max_{m=1,...,M} \|y_m\|. \quad (28)$$

Then, in the one-dimensional sum in (5), it holds that $\langle x_n - y_m, \xi \rangle \leq R$ for all $\xi \in \mathbb{S}^{d-1}$. Therefore, we can replace $f(|\cdot|)$ in (5) by any function $g \colon \mathbb{R} \to \mathbb{R}$ that satisfies $g(t) = f(|t|)$ for all $|t| \leq R$ without changing the sum. In particular, we choose $g$ as a sufficiently smooth $2T$-periodic function for some $T > R$, in order to achieve a convergent Fourier series of $g$. With this, we insert the one-dimensional summation (26) with a $2T$-periodic function $g$ into the sliced kernel sum (5) and obtain

$$\tilde{s}_m^{\text{FS}} := \frac{1}{P} \sum_{p=1}^{P} \sum_{n=1}^{N} \sum_{k=-N_{\text{ft}}/2}^{N_{\text{ft}}/2-1} w_n c_k(g) \mathrm{e}^{\pi \mathrm{i} k \langle y_m - x_n, \xi_p \rangle / T} \quad (29)$$

with the Fourier coefficients $c_k(g) = (2T)^{-1} \int_{-T}^{T} g(t) \exp(-\pi \mathrm{i} k t / T) \, \mathrm{d}t$. As already noted in Appendix I, we interchange the sums in order to achieve a fast summation

$$\tilde{s}_m^{\text{FS}} = \frac{1}{P} \sum_{p=1}^{P} \sum_{k=-N_{\text{ft}}/2}^{N_{\text{ft}}/2-1} c_k(g) \mathrm{e}^{\pi \mathrm{i} k \langle y_m, \xi_p \rangle / T} \sum_{n=1}^{N} w_n \mathrm{e}^{-\pi \mathrm{i} k \langle x_n, \xi_p \rangle / T}, \qquad m = 1, \dots, M.$$

Utilizing the NFFT for the sums over $n$ and $k$, this has a complexity of

$$\mathcal{O}(P(N + M + N_{\text{ft}} \log N_{\text{ft}}))$$

with $P$ the number of slices and $N_{\text{ft}}$ the number of Fourier coefficients.

The asymptotic complexities with respect to $M, N$ of the two methods depend on different parameters, most notably the number $D$ of random features for RFF and the number $P$ of points/slices in the QMC sequence for Fourier slicing, which need to be chosen depending on the desired accuracy. The following proposition compares their complexity to achieve a relative error

$$\frac{1}{\|w\|_1}\mathbb{E}\left[\|s - \tilde{s}^{\text{RFF}}\|_\infty\right] \in \mathcal{O}(\varepsilon) \quad \text{and} \quad \frac{1}{\|w\|_1}\|s - \tilde{s}^{\text{FS}}\|_\infty \in \mathcal{O}(\varepsilon) \qquad \text{for } \varepsilon \downarrow 0,$$

where $\|w\|_1 := \sum_{n=1}^N |w_n|$ and $\|s\|_\infty := \max_{m=1,\dots,M} |s_m|$, and, in case of RFF, the expectation is taken with respect to the random $v_p \sim \mu$.

**Proposition 5.** *Let $F(t) = \exp(-\frac{t^2}{2\sigma^2})$ be the Gauss kernel with $\sigma > 0$ and sliced kernel $f$. Assume there exists $R > 0$ satisfying (28) independently of $N$ and $M$.*

- *The computation of the RFF sum $\tilde{s}^{\text{RFF}} = (\tilde{s}_m^{\text{RFF}})_{m=1}^M$, see (27), with the parameter $D \sim \varepsilon^2 |\log \varepsilon|$ for $\varepsilon \downarrow 0$ achieves the relative error $\mathbb{E}[\|s - \tilde{s}\|_\infty]/\|w\|_1 \in \mathcal{O}(\varepsilon)$ and the numerical complexity*

$$\mathcal{O}\big((N + M)\,\varepsilon^{-2}|\log \varepsilon|\big).$$

- *For $q \in \mathbb{N}$ arbitrary, let $g \in C^q(\mathbb{R}^d)$ be a $2T$-periodic function with $T > R$ that satisfies $g(t) = f(|t|)$ for $|t| \leq R$. Furthermore, assume that the slicing error has rate $r > 0$, i.e.,*

$$\sup_{x \in \mathbb{R}^d} \left| F(\|x\|) - \frac{1}{P}\sum_{p=1}^P f(|\langle \xi_p, x \rangle|) \right| \in \mathcal{O}(P^{-r}).$$

*Then the computation of the Fourier slicing sum $\tilde{s}^{\text{FS}} = (\tilde{s}_m^{\text{FS}})_{m=1}^M$, see (29), with the parameters $P \sim \varepsilon^{-1/r}$ and $N_{\text{ft}} \sim \varepsilon^{-1/q}|\log \varepsilon|^{-1}$ for $\varepsilon \downarrow 0$ achieves the relative error $\|s - \tilde{s}^{\text{FS}}\|_\infty/\|w\|_1 \in \mathcal{O}(\varepsilon)$ and the numerical complexity*

$$\mathcal{O}\big(\varepsilon^{-1/r}(\varepsilon^{-1/q} + N + M)\big).$$

The assumption that the slicing error is bounded with rate $r$ is fulfilled with $r = \frac{d}{2(d-1)} > \frac{1}{2}$ for the distance QMC designs, which minimize (9) with $s = d/2$, as we proved in Corollary 4. However, our numerical results suggest that $r$ might be even larger, cf. Table 1. Because $q \in \mathbb{N}$ can be chosen arbitrarily and $r > 1/2$, the asymptotic complexity of the slicing summation is lower than for RFF. Furthermore, we note that the second part of the proposition holds for any kernel function $F$ for which $t \mapsto f(|t|)$ is in $C^q(\mathbb{R})$.

*Proof.* **RFF:** By (Sutherland & Schneider, 2015, Prop 3), we have for the Gauss kernel

$$\mathbb{E}_{v_1,\dots,v_D \sim \mu}\left[\sup_{x \in B_T}\left|F(\|\cdot\|) - \frac{1}{D}\sum_{p=1}^D e^{-2\pi i \langle \cdot, v_p\rangle}\right|\right]$$

$$\leq \frac{24\sqrt{d}T(e^{-1/2} + \sqrt{d} + \sqrt{2\log(D)})}{\sigma\sqrt{D}} \in \mathcal{O}\left(\sqrt{\frac{\log(D)}{D}}\right),$$

where $B_T$ the ball in $\mathbb{R}^d$ of radius $T > 0$. Hence, the error of the RFF summation is bounded by

$$\mathbb{E}[\|s - \tilde{s}^{\text{RFF}}\|_\infty] \leq \sum_{n=1}^N |w_n| \mathbb{E}\left[\max_{m=1,\dots,M}\left|F(\|x_n - y_m\|) - \frac{1}{D}\sum_{p=1}^D e^{2\pi i \langle y_m - x_n, v_p\rangle}\right|\right]$$

$$\leq \|w\|_1 \mathbb{E}\left[\max_{\substack{n=1,\dots,N \\ m=1,\dots,M}}\left|F(\|x_n - y_m\|) - \frac{1}{D}\sum_{p=1}^D e^{2\pi i \langle y_m - x_n, v_p\rangle}\right|\right] \in \mathcal{O}\left(\sqrt{\frac{\log(D)}{D}}\right).$$

For the desired accuracy $\varepsilon$, we choose $D \sim -\varepsilon^{-2}\log(\varepsilon)$. Then $\frac{\log(D)}{D} = \varepsilon^2 \frac{\log(-\log(\varepsilon)) - 2\log(\varepsilon)}{-\log(\varepsilon)} \in \mathcal{O}(\varepsilon^2)$, so that we obtain a relative error $\mathcal{O}(\varepsilon)$ with a complexity of

$$\mathcal{O}\left(D(N + M)\right) = \mathcal{O}\left((N + M)\,\varepsilon^{-2}|\log(\varepsilon)|\right).$$

**Slicing:** For the Gauss kernel, the sliced kernel $t \mapsto f(|t|)$ is an analytic function, see Table 2. Therefore, $g$ can be constructed via two-point Taylor approximation, see Potts & Steidl (2003). We estimate the error

$$\|s - \tilde{s}^{\mathrm{FS}}\|_\infty \leq \|w\|_1 \max_{\substack{n=1,\ldots,N \\ m=1,\ldots,M}} \left| F(\|x_n - y_m\|) - \frac{1}{P} \sum_{p=1}^{P} \sum_{k=-N_{\mathrm{ft}}/2}^{N_{\mathrm{ft}}/2-1} c_k(g) \mathrm{e}^{\pi \mathrm{i} k \langle y_m - x_n, \xi_p \rangle / T} \right|.$$

Since, $R > \|x_n - y_m\|$ for all $n$ and $m$ by (28), we have

$$\frac{\|s - \tilde{s}^{\mathrm{FS}}\|_\infty}{\|w\|_1} \leq \sup_{\|x\| \leq R} \left| F(\|x\|) - \frac{1}{P} \sum_{p=1}^{P} f(|\langle \xi_p, x \rangle|) \right|$$

$$+ \sup_{\|x\| \leq R} \left| \frac{1}{P} \sum_{p=1}^{P} \left( f(|\langle \xi_p, x \rangle|) - \sum_{k=-N_{\mathrm{ft}}/2}^{N_{\mathrm{ft}}/2-1} c_k(g) \mathrm{e}^{\pi \mathrm{i} k \langle x, \xi_p \rangle / T} \right) \right|.$$

Since $g(t) = f(|t|)$ for all $|t| \leq R$, we have

$$\frac{\|s - \tilde{s}^{\mathrm{FS}}\|_\infty}{\|w\|_1} \leq \sup_{x \in \mathbb{R}^d} \left| F(\|x\|) - \frac{1}{P} \sum_{p=1}^{P} f(|\langle \xi_p, x \rangle|) \right| + \sup_{|t| \leq R} \left| g(t) - \sum_{k=-N_{\mathrm{ft}}/2}^{N_{\mathrm{ft}}/2-1} c_k(g) \mathrm{e}^{\pi \mathrm{i} k t / T} \right|.$$

For the first term, we use our assumption that

$$\sup_{x \in \mathbb{R}^d} \left| F(\|x\|) - \frac{1}{P} \sum_{p=1}^{P} f(|\langle \xi_p, x \rangle|) \right| \in \mathcal{O}(P^{-r})$$

with some $r > 0$. For the second term, Bernstein's theorem for the convergence of Fourier series implies that there exists $C_g$ such that

$$\sup_{t \in \mathbb{R}} \left| g(t) - \sum_{k=-N_{\mathrm{ft}}/2}^{N_{\mathrm{ft}}/2-1} c_k(g) \mathrm{e}^{\pi \mathrm{i} k t / T} \right| \leq C_g N_{\mathrm{ft}}^{-q} \log(N_{\mathrm{ft}}),$$

cf. (Plonka et al., 2023, Thm 1.39). Combining these estimates, we obtain

$$\frac{\|s - \tilde{s}^{\mathrm{FS}}\|_\infty}{\|w\|_1} \in \mathcal{O}(P^{-r} + N_{\mathrm{ft}}^{-q} \log(N_{\mathrm{ft}})).$$

Choosing $P \sim \varepsilon^{-1/r}$ and $N_{\mathrm{ft}} \sim \varepsilon^{-1/q} / \log(\varepsilon^{-1})$ for $\varepsilon \downarrow 0$, we see that

$$P^{-r} + N_{\mathrm{ft}}^{-q} \log(N_{\mathrm{ft}}) \sim \varepsilon + \frac{\varepsilon(q^{-1} \log(\varepsilon^{-1}) - \log(\log(\varepsilon^{-1})))}{\log(\varepsilon^{-1})^q} \in \mathcal{O}(\varepsilon).$$

Hence, the complexity of the Fourier slicing to achieve relative error $\varepsilon$ is

$$\mathcal{O}(P(N + M + N_{\mathrm{ft}} \log N_{\mathrm{ft}})) = \mathcal{O}\big(\varepsilon^{-1/r}(\varepsilon^{-1/q} + N + M)\big). \quad \square$$

## K  ADDITIONAL NUMERICAL RESULTS

### K.1  ADDITIONAL PLOTS AND TABLES FOR SECTION 4.2

In the following, we redo the experiments from Figure 1 and Table 1 and vary some parameters. More precisely, we redo it for the negative distance kernel, choose other length scale parameters of the kernel and perform it in higher dimensions ($d = 200$).

**Negative Distance Kernel**  We do the same experiment as in Figure 1 and Table 1 with the negative distance kernel. The results are given in Figure 4 and Table 3. We can see that the advantage of QMC slicing is not as large as for smooth kernels, which is expected considering the theoretical results from Section 3. In particular, the spherical function (10) is not in $H^{d/2}(\mathbb{S}^{d-1})$ if $d \geq 3/2$, so the assumptions of the bound (8) are not fulfilled. Nevertheless, QMC slicing is still significantly more

accurate than non-QMC slicing. Note that RFF based methods are not available for the negative distance kernel since it is not positive definite and therefore Bochner's theorem does not apply.

**Other Length Scales of the Kernels**  We redo the experiment from Figure 1 and Table 1 with scale factors $s = \frac{1}{2}$ and $s = 2$ of the kernel parameter. The results are given in Figure 5 and Table 4 for $s = \frac{1}{2}$ and in Figure 6 and Table 5 for $s = 2$. We observe that the advantage of QMC is more significant of for larger scale factors, which is expected since the function $\xi \mapsto f(|\langle \xi, x \rangle|)$ is more regular for larger $s$ than for smaller $s$.

**Higher Dimensions**  Finally, we do the same experiment as in Figure 1 and Table 1 for the higher dimension $d = 200$. Here, we use the negative distance kernel, the Matérn kernel with $\nu = 3 + \frac{1}{2}$ and the Gauss kernel, where the parameters are chosen by the median rule with scale factor $\gamma = 1$. The results are given in Figure 7 and Table 6. The advantage of QMC is less pronounced in such high dimensions, but still visible.

## K.2  ADDITIONAL RESULTS FOR SECTION 4.3

**Negative Distance Kernel**  We redo the experiment from Section 4.3 for the negative distance kernel and the thin-plate spline kernel. The results are given in Figure 8. We can see that QMC Fourier slicing outperforms standard slicing clearly in all cases. Note that RFF based methods are not available for these kernels since they are not positive definite and Bochner's theorem does not apply.

**Higher Dimensions**  We run the same experiment as in Section 4.3 on the MNIST and FashionM-NIST dataset without dimension reduction and therefore $d = 784$. The results are given in Figure 9. We can see that the advantage of QMC slicing is smaller than for the lower-dimensional examples but still clearly visible for some kernels. In accordance with the considerations of Appendix H, the advantage comes in when $P > d$.

**GPU Comparison**  We want to demonstrate the advantage of our method in a GPU-comparison with a large number of data points. As a test dataset we concatenate the MNIST and FashionMNIST in all eight orientations arising rotating and mirroring the images and reduce the dimension via PCA to $d = 30$. The arising dataset has $N = M = 960000$ entries. Then, we compare RFF, ORF, QMC (Sobol) RFF, Slicing and QMC Slicing, where the QMC directions for slicing are chosen by minimizing the distance functional, see Section 4.1. This experiment is implemented in Python using PyTorch and we use brute-force kernel summation by the PyKeOps library (Charlier et al., 2021) as a baseline. The results are given in Figure 10. Even though the RFF-based methods parallelize a bit better on the GPU than the fast Fourier summations, the conclusions are mainly the same as for the CPU experiments. We can clearly see the advantage of QMC slicing over the comparisons. Particularly, for non-smooth kernels, slicing-based methods work much better than RFF-based methods.

## K.3  MMD GRADIENT FLOWS

Finally, we use our fast summation method in a specific application. Here, we consider gradient flows of the maximum mean discrepancy (MMD), which have been considered in several papers for generative modeling and other applications, see, e.g., Arbel et al. (2019); Chen et al. (2024); Galashov et al. (2024); Hertrich et al. (2024); Lim et al. (2024).

**Background**  For a dataset $\boldsymbol{y} = (y_1, ..., y_M)$ of target particles, we consider the discrete MMD functional $F_{\boldsymbol{y}} : (\mathbb{R}^d)^N \to \mathbb{R}$ defined by

$$G_{\boldsymbol{y}}(\boldsymbol{x}) = \mathrm{MMD}_K(\boldsymbol{x}, \boldsymbol{y}) = \frac{1}{2N^2} \sum_{i,j=1}^{N} K(x_i, x_j) - \frac{1}{MN} \sum_{i,j=1}^{N,M} K(x_i, y_j) + \frac{1}{2M^2} \sum_{i,j=1}^{M} K(y_i, y_j).$$

We note that (under suitable assumptions on the kernel), the MMD is a metric on the space of probability distributions on $\mathbb{R}^d$ such that $F_{\boldsymbol{y}}(\boldsymbol{x})$ is always non-negative and zero if and only if the particles $\boldsymbol{x}$ and $\boldsymbol{y}$ coincide. Here, we interpret $\boldsymbol{x}$ as the discrete probability measure $\frac{1}{N} \sum_{n=1}^{N} \delta_{x_n}$.

Now, we minimize $G_{\boldsymbol{y}}$ by simulating the gradient flow

$$\dot{\boldsymbol{x}} = -\nabla G_{\boldsymbol{y}}(\boldsymbol{x}), \quad \boldsymbol{x}(0) = \boldsymbol{x}^{(0)},$$

starting with some random initial samples $\boldsymbol{x}^{(0)} \in (\mathbb{R}^d)^N$ using the explicit Euler discretization

$$\boldsymbol{x}^{(k+1)} = \boldsymbol{x}^{(k)} - \tau \nabla G_{\boldsymbol{y}}(\boldsymbol{x}^{(k)}), \tag{30}$$

where the gradient $\nabla G_{\boldsymbol{y}}$ is computed via backpropagation. Throughout the flow, the particles $\boldsymbol{x}$ will then move towards the target distribution $\boldsymbol{y}$.

**Experimental Setup** We choose the negative distance kernel $K(x, y) = -\|x - y\|$ and consider the CIFAR10 dataset ($M = 50000$ and $d = 3072$) as target points $\boldsymbol{y}$. For $\boldsymbol{x}$, we choose $N = M = 50000$ samples. Then, we run the (discretized) gradient flow from (30), with initial particles $\boldsymbol{x}^{(0)}$ drawn iid from a standard normal distribution. For speeding up the convergence, we follow Hertrich et al. (2024); Lim et al. (2024) and add a momentum parameter $m = 0.9$. That is, we modify the equation (30) to

$$\boldsymbol{v}^{(k+1)} = \nabla G_{\boldsymbol{y}}(\boldsymbol{x}^{(k)}) + m\boldsymbol{v}^{(k)}$$
$$\boldsymbol{x}^{(k+1)} = \boldsymbol{x}^{(k)} - \tau \boldsymbol{v}^{(k+1)}$$

with initial value $\boldsymbol{v}^{(0)} = 0$. We run the MMD flow for $50000$ steps with step size $\tau = 1$, where we compute the function $G_{\boldsymbol{y}}$ by (Monte Carlo) Slicing and by QMC Slicing with $P = 1000$ projections.

**Remark 6.** *We would like to point out that running the gradient flow with exact computation of $G_{\boldsymbol{y}}$ is computationally intractable. Using (QMC-)Slicing, one iteration (30) takes between $0.2$ and $0.3$ seconds on an NVIDIA RTX 4090 GPU. On the other hand, the exact gradient evaluation via PyKeOps takes about one hour. Considering that we are running the flow for $50000$ steps, this underlines the need of (QMC-)Slicing.*

**Results** We plot the objective value of $G_{\boldsymbol{y}}(\boldsymbol{x})$ versus the computation time in Figure 11. We observe that the smaller error in the gradient evaluation by QMC Slicing significantly improves the convergence behavior.

### K.4 ON THE GAP BETWEEN THEORETICAL GUARANTEES AND NUMERICAL RESULTS

In our numerical part, we observe significantly better error rates than we can prove theoretically. One possible explanation is the following. Our theoretical guarantees for QMC Slicing are based on worst-case errors in Sobolev spaces on the sphere and consequently rely on the smoothness of the functions $g_x$ from (10), which is not satisfied for some kernels in Theorem 3. However, these results are only worst-case error rates that do not account for the specific properties of $g_x$. First, the function $g_x(\xi)$ depends only on $\langle x, \xi \rangle$, thus having a lower effective dimension. Furthermore, by construction $g_x(\xi) = g_x(-\xi)$ such that for all $x \in \mathbb{R}^d$ it holds that $\mathbb{E}_{\xi \sim \mathcal{U}_{\mathbb{S}^{d-1}}}[g_x(\xi)] = \mathbb{E}_{\xi \sim \mathcal{U}_{\{\zeta \in \mathbb{S}^{d-1} : \langle \zeta, x \rangle > 0\}}}[g_x(\xi)]$. Moreover, $g_x$ is infinitely often differentiable for on the hemisphere $\{\zeta \in \mathbb{S}^{d-1} : \langle \zeta, x \rangle > 0\}$ for all considered kernels of Appendix A such that tighter error bounds could apply. Consequently, exploring QMC designs on the hemisphere could be an interesting direction for further improving our theoretical analysis.

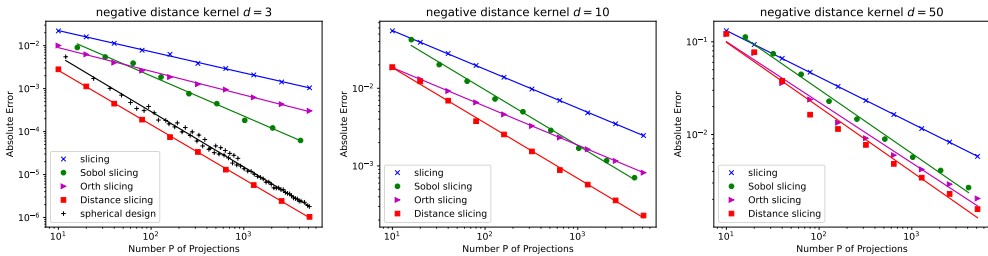

Figure 4: Loglog plot of the approximation error in (4) versus the number $P$ of projections for the negative distance kernel. The results are averaged over 50 realizations of $\boldsymbol{\xi}^P$ and 1000 realizations of $x$. We fit a regression line in the loglog plot for each method to estimate the convergence rate, see also Table 3.

Table 3: Estimated convergence rates for the different methods and the negative distance kernel. We estimate the rate $r$ by fitting a regression line in the loglog plot. Then, we obtain the estimated convergence rate $\mathcal{O}(P^{-r})$ for some $r > 0$. Consequently, larger values of $r$ correspond to a faster convergence. The resulting values of $r$ are given in the below tables, the best values are highlighted in bolt. See Figure 4 for a visualization.

| | Negative distance kernel | | | | |
| --- | --- | --- | --- | --- | --- |
| | Slicing-based | | | | |
| Dimension | Slicing | Sobol | Orth | Distance | spherical design |
| $d = 3$ | 0.49 | 0.94 | 0.55 | 1.27 | **1.29** |
| $d = 10$ | 0.50 | 0.72 | 0.50 | **0.71** | - |
| $d = 50$ | 0.50 | 0.69 | 0.65 | **0.70** | - |

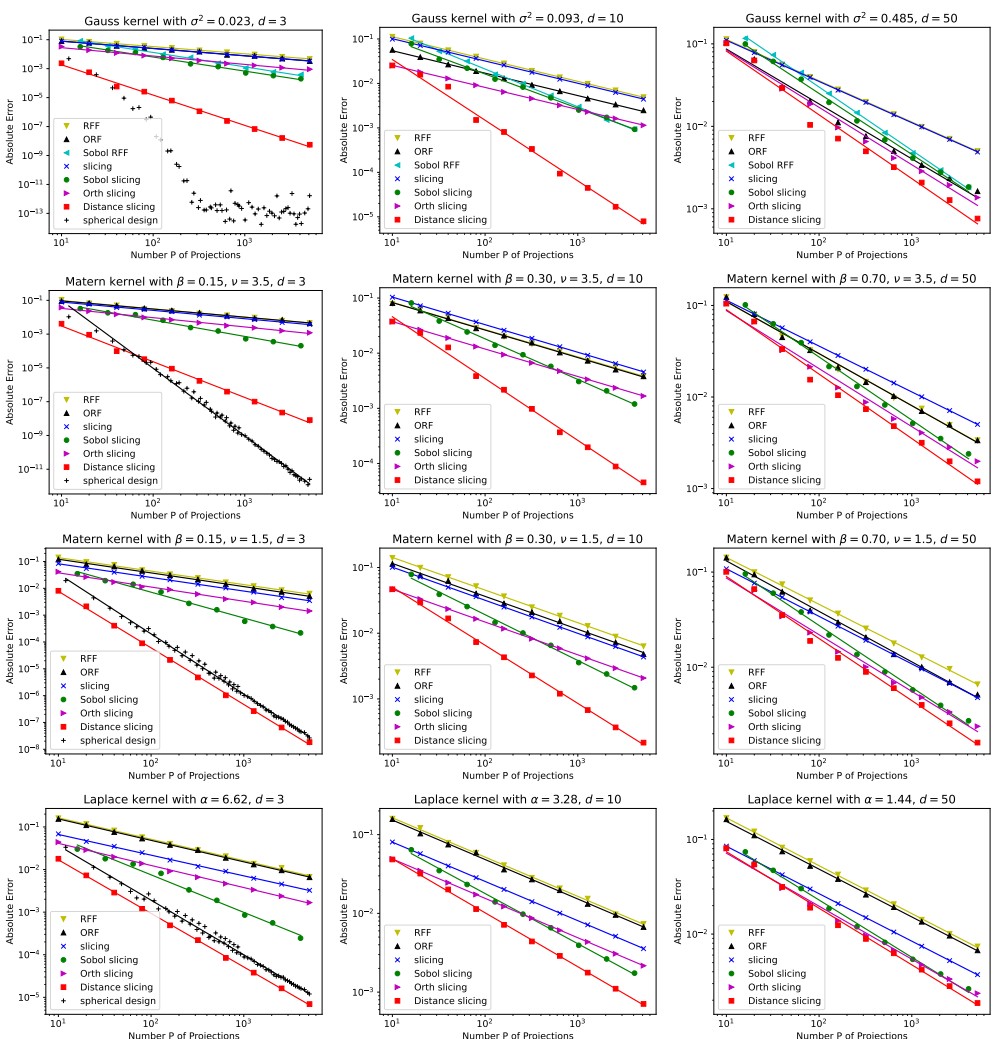

Figure 5: Loglog plot of the approximation error in (4) versus the number $P$ of projections for different kernels and dimensions (left $d = 3$, middle $d = 10$, right $d = 50$). The results are averaged over 50 realizations of $\boldsymbol{\xi}^P$ and 1000 realizations of $x$. The kernel parameters are set by the median rule with scale factor $\gamma = \frac{1}{2}$. We fit a regression line in the loglog plot for each method to estimate the convergence rate, see also Table 4.

Table 4: Estimated convergence rates for the different methods. We estimate the rate $r$ by fitting a regression line in the loglog plot. Then, we obtain the estimated convergence rate $\mathcal{O}(P^{-r})$ for some $r > 0$. Consequently, larger values of $r$ correspond to a faster convergence. The resulting values of $r$ are given in the below tables, the best values are highlighted in bolt. The kernel parameters are the same as in Figure 5 (scale factor $\gamma = \frac{1}{2}$).

| Gauss kernel with kernel scaling $\gamma = \frac{1}{2}$ | | | | | | | |
|---|---|---|---|---|---|---|---|
| | RFF-based | | | Slicing-based | | | |
| Dimension | RFF | Sobol | ORF | Slicing | Sobol | Orth | Distance |
| $d = 3$ | 0.50 | 0.99 | 0.50 | 0.51 | 0.97 | 0.58 | **2.09** |
| $d = 10$ | 0.50 | 0.85 | 0.50 | 0.50 | 0.77 | 0.50 | **1.36** |
| $d = 50$ | 0.50 | **0.77** | 0.67 | 0.50 | 0.74 | 0.70 | **0.77** |

| Matérn kernel with $\nu = 3 + \frac{1}{2}$ and kernel scaling $\gamma = \frac{1}{2}$ | | | | | | | |
|---|---|---|---|---|---|---|---|
| | RFF-based | | Slicing-based | | | | |
| Dimension | RFF | ORF | Slicing | Sobol | Orth | Distance | spherical design |
| $d = 3$ | 0.49 | 0.48 | 0.49 | 0.96 | 0.55 | 2.11 | **4.01** |
| $d = 10$ | 0.49 | 0.50 | 0.50 | 0.74 | 0.50 | **1.12** | - |
| $d = 50$ | 0.57 | 0.57 | 0.50 | 0.69 | 0.63 | **0.70** | - |

| Matérn kernel with $\nu = 1 + \frac{1}{2}$ and kernel scaling $\gamma = \frac{1}{2}$ | | | | | | | |
|---|---|---|---|---|---|---|---|
| | RFF-based | | Slicing-based | | | | |
| Dimension | RFF | ORF | Slicing | Sobol | Orth | Distance | spherical design |
| $d = 3$ | 0.50 | 0.51 | 0.51 | 0.96 | 0.53 | 2.11 | **2.24** |
| $d = 10$ | 0.50 | 0.50 | 0.50 | 0.70 | 0.50 | **0.88** | - |
| $d = 50$ | 0.49 | 0.53 | 0.50 | **0.65** | 0.60 | **0.65** | - |

| Laplace kernel with kernel scaling $\gamma = \frac{1}{2}$ | | | | | | | |
|---|---|---|---|---|---|---|---|
| | RFF-based | | Slicing-based | | | | |
| Dimension | RFF | ORF | Slicing | Sobol | Orth | Distance | spherical design |
| $d = 3$ | 0.50 | 0.50 | 0.49 | 0.88 | 0.52 | 1.26 | **1.28** |
| $d = 10$ | 0.50 | 0.50 | 0.50 | 0.64 | 0.50 | **0.69** | - |
| $d = 50$ | 0.51 | 0.50 | 0.50 | **0.61** | 0.56 | 0.60 | - |

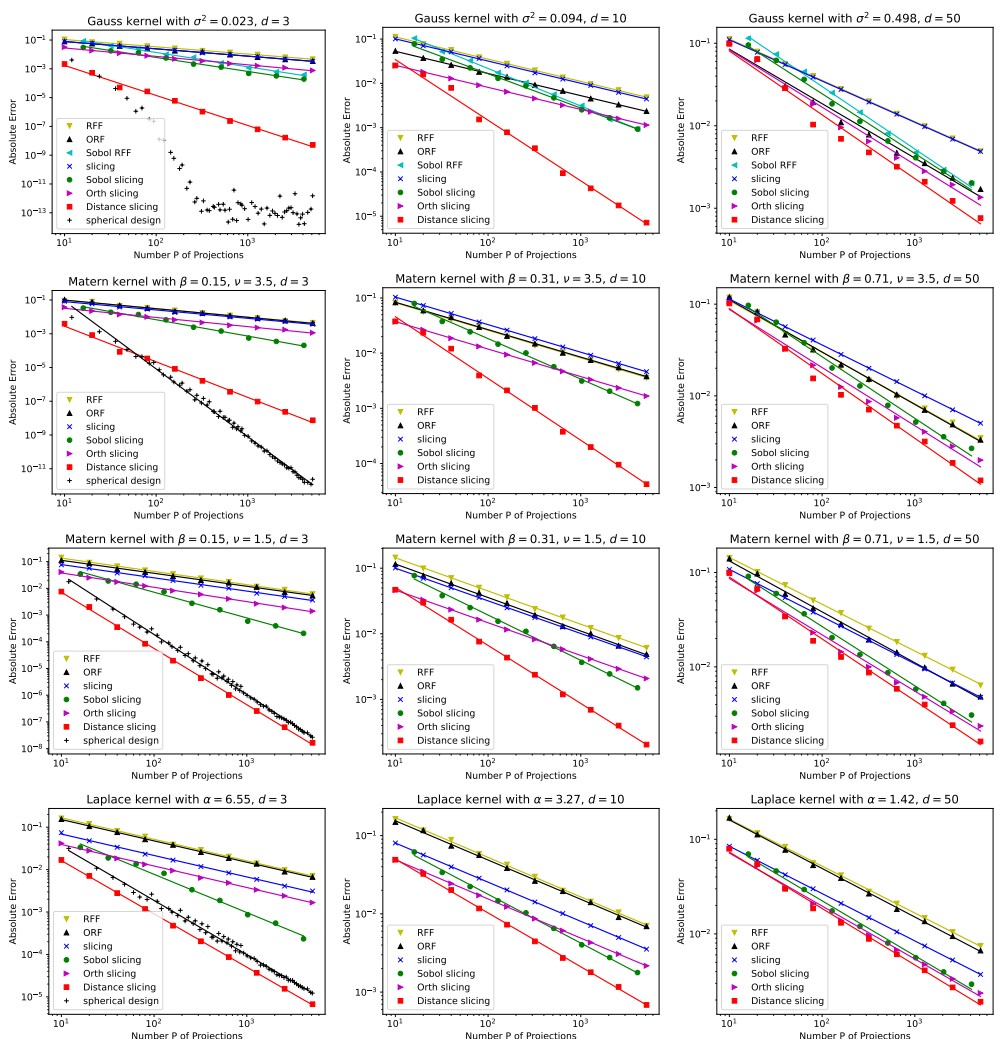

Figure 6: Loglog plot of the approximation error in (4) versus the number $P$ of projections for different kernels and dimensions (left $d = 3$, middle $d = 10$, right $d = 50$). The results are averaged over 50 realizations of $\boldsymbol{\xi}^P$ and 1000 realizations of $x$. The kernel parameters are set by the median rule with scale factor $\gamma = 2$. We fit a regression line in the loglog plot for each method to estimate the convergence rate, see also Table 5.

Table 5: Estimated convergence rates for the different methods. We estimate the rate $r$ by fitting a regression line in the loglog plot. Then, we obtain the estimated convergence rate $\mathcal{O}(P^{-r})$ for some $r > 0$. Consequently, larger values of $r$ correspond to a faster convergence. The resulting values of $r$ are given in the below tables, the best values are highlighted in bolt. The kernel parameters are the same as in Figure 6 (scale factor $\gamma = 2$).

| Gauss kernel with kernel scaling $\gamma = 2$ | | | | | | | |
| --- | --- | --- | --- | --- | --- | --- | --- |
| | RFF-based | | | Slicing-based | | | |
| Dimension | RFF | Sobol | ORF | Slicing | Sobol | Orth | Distance |
| $d = 3$ | 0.50 | 1.00 | 0.51 | 0.51 | 0.97 | 0.59 | **2.08** |
| $d = 10$ | 0.50 | 0.85 | 0.50 | 0.50 | 0.77 | 0.50 | **1.37** |
| $d = 50$ | 0.50 | 0.76 | 0.66 | 0.50 | 0.72 | 0.70 | **0.77** |

| Matérn kernel with $\nu = 3 + \frac{1}{2}$ and kernel scaling $\gamma = 2$ | | | | | | | |
| --- | --- | --- | --- | --- | --- | --- | --- |
| | RFF-based | | Slicing-based | | | | |
| Dimension | RFF | ORF | Slicing | Sobol | Orth | Distance | spherical design |
| $d = 3$ | 0.50 | 0.51 | 0.50 | 0.97 | 0.54 | 2.11 | **4.02** |
| $d = 10$ | 0.50 | 0.50 | 0.50 | 0.73 | 0.50 | **1.12** | - |
| $d = 50$ | 0.56 | 0.57 | 0.50 | 0.67 | 0.63 | **0.71** | - |

| Matérn kernel with $\nu = 1 + \frac{1}{2}$ and kernel scaling $\gamma = 2$ | | | | | | | |
| --- | --- | --- | --- | --- | --- | --- | --- |
| | RFF-based | | Slicing-based | | | | |
| Dimension | RFF | ORF | Slicing | Sobol | Orth | Distance | spherical design |
| $d = 3$ | 0.50 | 0.49 | 0.50 | 0.96 | 0.54 | 2.10 | **2.24** |
| $d = 10$ | 0.51 | 0.51 | 0.50 | 0.69 | 0.50 | **0.88** | - |
| $d = 50$ | 0.50 | 0.54 | 0.50 | 0.63 | 0.60 | **0.65** | - |

| Laplace kernel with kernel scaling $\gamma = 2$ | | | | | | | |
| --- | --- | --- | --- | --- | --- | --- | --- |
| | RFF-based | | Slicing-based | | | | |
| Dimension | RFF | ORF | Slicing | Sobol | Orth | Distance | spherical design |
| $d = 3$ | 0.51 | 0.50 | 0.51 | 0.90 | 0.51 | 1.26 | **1.28** |
| $d = 10$ | 0.51 | 0.50 | 0.50 | 0.62 | 0.50 | **0.69** | - |
| $d = 50$ | 0.50 | 0.51 | 0.50 | 0.58 | 0.56 | **0.60** | - |

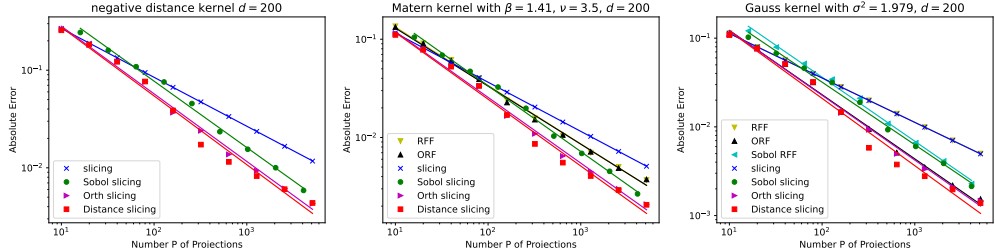

Figure 7: Loglog plot of the approximation error in (4) versus the number $P$ of projections for $d = 200$. The results are averaged over $50$ realizations of $\boldsymbol{\xi}^P$ and $1000$ realizations of $x$. The kernel parameters are chosen by the median rule with scale factor $\gamma = 1$. We fit a regression line in the loglog plot for each method to estimate the convergence rate, see also Table 6.

Table 6: Estimated convergence rates for the different methods and kernels for $d = 200$. We estimate the rate $r$ by fitting a regression line in the loglog plot. Then, we obtain the estimated convergence rate $\mathcal{O}(P^{-r})$ for some $r > 0$. Consequently, larger values of $r$ correspond to a faster convergence. The resulting values of $r$ are given in the below tables, the best values are highlighted in bolt. The kernel parameters are the same as in Figure 7 (scale factor $\gamma = 1$).

| Dimension $d = 200$ | | | | | | | |
|---|---|---|---|---|---|---|---|
| | RFF-based | | | Slicing-based | | | |
| Kernel | RFF | Sobol | ORF | Slicing | Sobol | Orth | Distance |
| Negative Distance | - | - | - | 0.50 | 0.68 | 0.69 | **0.70** |
| Matérn, $\nu = 3 + \frac{1}{2}$ | 0.60 | - | 0.59 | 0.50 | 0.67 | 0.67 | **0.68** |
| Gauss | 0.50 | 0.72 | 0.72 | 0.50 | 0.71 | 0.73 | **0.76** |

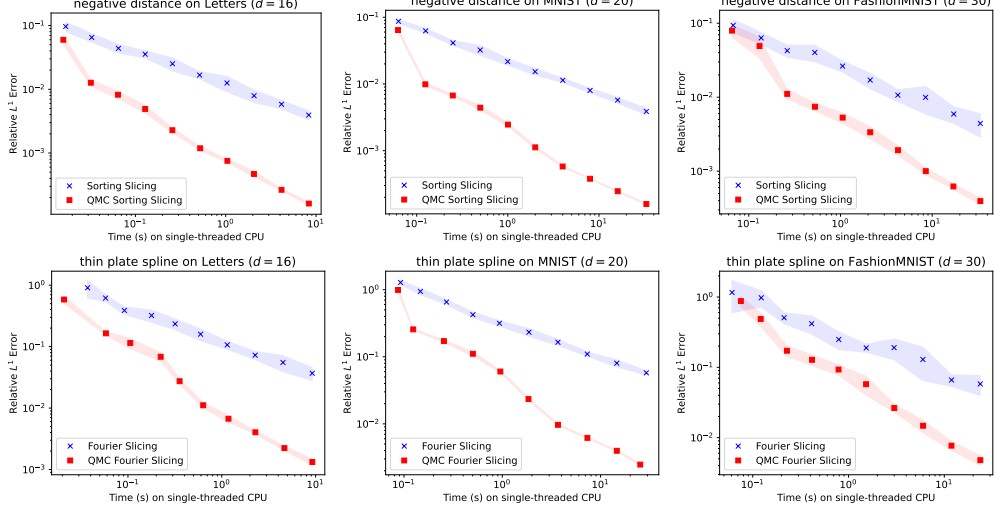

Figure 8: Loglog plot of the relative $L^1$ approximation error versus computation time for computing the kernel summations (1) with the negative distance and thin plate spline kernel and different methods and datasets. MNIST and FashionMNIST are reduced to dimension $d = 20$ and $d = 30$ via PCA. We run each method 10 times. The shaded area indicates the standard deviation of the error. For the slicing method, we use $P = 10 \cdot 2^k$ slices for $k = 0, ..., 9$.

MNIST

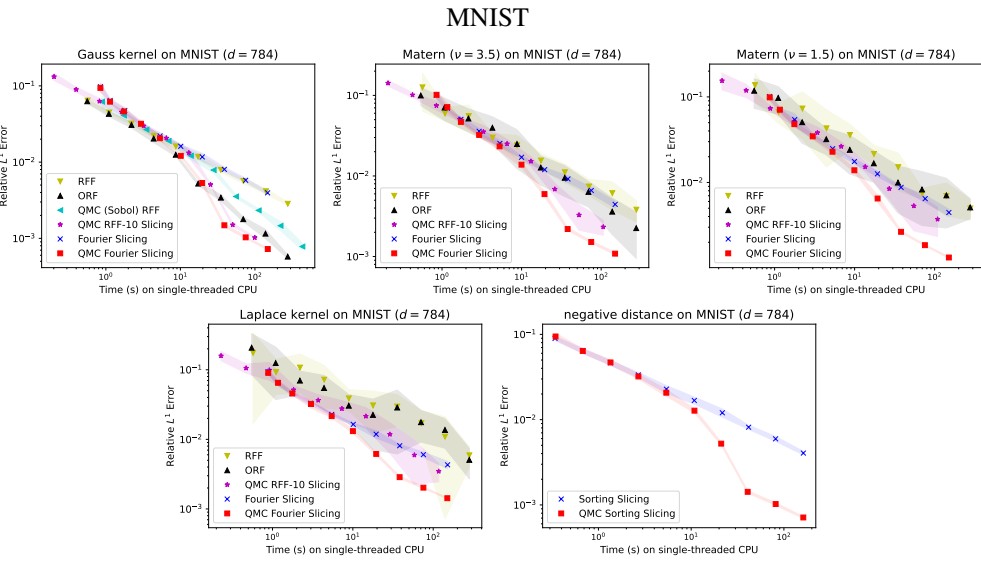

FashionMNIST

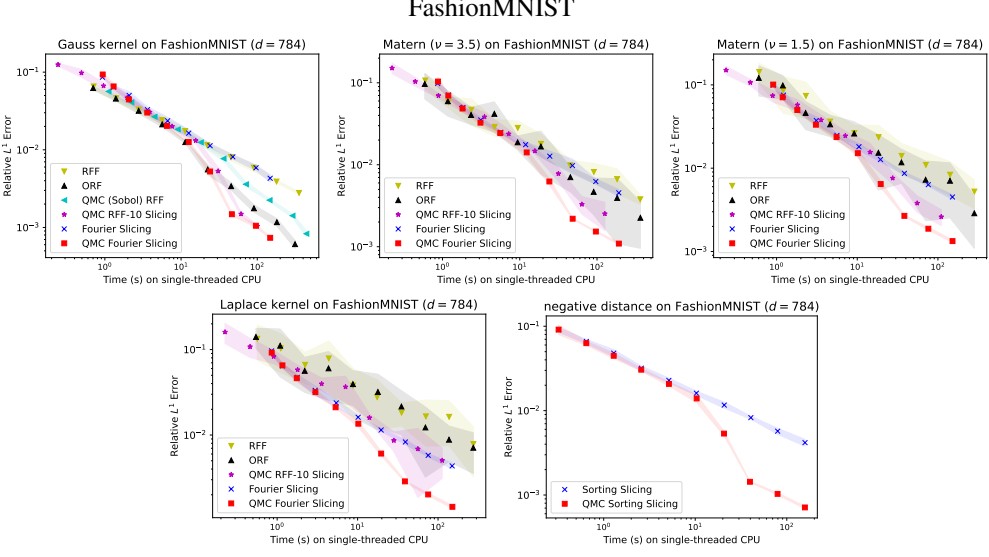

Figure 9: Loglog plot of the relative $L^1$ approximation error versus computation time for computing the kernel summations (1) with different kernels and methods on the MNIST and FashionMNIST dataset without dimension reduction. We run each method 10 times. The shaded area indicates the standard deviation of the error. For the slicing method, we use $P = 10 \cdot 2^k$ slices for $k = 0, ..., 9$. In order to obtain similar computation times, we run RFF and ORF with $D = 2P$ features.

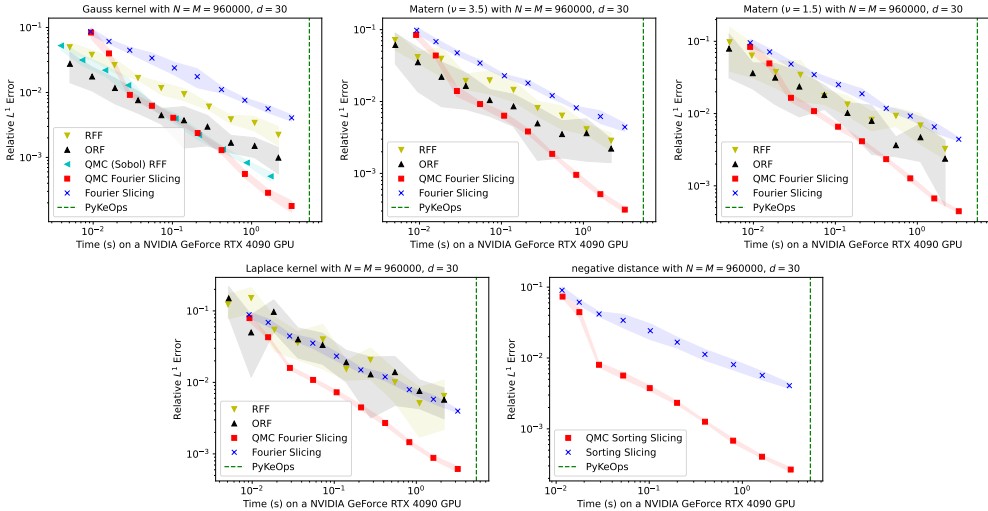

Figure 10: Loglog plot of the relative $L^1$ approximation error versus GPU computation time for computing the kernel summations (1) with different kernels on a large dataset ($M = N = 960000$). We run each method 10 times. The shaded area indicates the standard deviation of the error. For the slicing method, we use $P = 10 \cdot 2^k$ slices for $k = 0, ..., 9$. In order to obtain similar computation times, we run RFF and ORF with $D = 4P$ features. Since PyKeOps computes the exact kernel sum, the computation time is indicated by a vertical line.

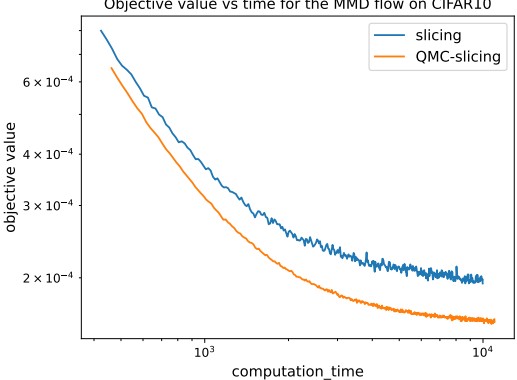

Figure 11: Plot of the objective value $F_{\boldsymbol{y}}(\boldsymbol{x})$ for the MMD gradient flow on the CIFAR10 dataset on an NVIDIA GeForce RTX 4090 GPU. Note that computing this flow with exact kernel summations is computationally intractable. The plot omits the first 2000 steps of the gradient flow to improve the scaling of the plot.

