# OpenReview forum: "Fast Summation of Radial Kernels via QMC Slicing"
_ICLR.cc/2025/Conference — ICLR 2025 Poster_

### Official Review · Reviewer_Edek · 2024-11-01

**Soundness:** 3
**Presentation:** 3
**Contribution:** 3
**Rating:** 8
**Confidence:** 3

**Summary:**

This paper derived the error bounds of slicing, a fast approximation algorithm for computing kernel sums, for various kernels. The paper also incorporated quasi-Monte Carlo (QMC) sequences on the sphere into the slicing method to improve its error rate. The paper conducted experiments to demonstrate that the QMC-slicing approach outperforms the non-QMC slicing method and other baselines like Random Fourier Features (RFF) in terms of error rate.

**Strengths:**

1. The authors proposed the QMC-slicing method and demonstrated that it has better error rates than the original slicing method. The QMC-slicing method exhibits a better expected error bound than RFF while maintaining a similar time complexity; this demonstrates its effectiveness and potential as a fast kernel summation algorithm.
2. The authors provided a comprehensive analysis of the slicing error for the negative distance kernel, the thin plate spline, the Laplace kernel and the Gauss kernel. The derived error bounds could be useful for future research on slicing and fast kernel summation.
3. Overall, the paper is well-written and clear to understand. Background knowledge on fast kernel summation, slicing and QMC are well-explained. Details of derivation, e.g. the applicability of the QMC methods for slicing, have been sufficiently provided.

**Weaknesses:**

1. The authors considered several QMC sequences in the experiments, and observed difference in slicing error between these different sequences. It would be helpful if the authors could further analyze the results and discuss more on how the choice of QMC sequences affects slicing error, under what circumstances would some QMC sequences outperform others, etc.
2. As mentioned in Section 5, the observed error rates of QMC-slicing are significantly better than the theory. It would be helpful if the authors discuss more about this phenomenon and the reason behind it.

**Questions:**

1. The authors mentioned that QMC-slicing exhibits a smaller advantage in high dimensions (d=200), but it still outperforms other baselines. What if the dimensionality is even higher, e.g. d=10,000? Does QMC-slicing still perform better than slicing and (QMC-)RFF?
2. In this paper, only Gauss, Laplace, Matern and negative distance kernels are considered. How would QMC-slicing perform in the case of other kernels, e.g. sigmoid kernel?

---

> ### Author Response · Authors · 2024-11-19
> **Response to the Reviewer**
>
> We would like to thank the reviewer for the evaluation of our paper. Please find below the detailed answers to the comments and questions.
>
> - Regarding Weakness 1:
> In our numerical experiments, the distance QMC designs perform best among the (computable) QMC rules. The only exception is when the (provably optimal) spherical designs are available, which are only computable in $d\le4$.
> This is now written clearly in Section 4.2.
> For high dimension $d$ and few directions $P<d$, we showed in Appendix H that orthogonal points and distance QMC designs coincide, but the latter yield better errors for more directions $P>d$.
>
> - Regarding Weakness 2:
> One possible explanation for this gap could be that our theoretical bounds on the error of QMC designs is based on worst-case error estimates in the Sobolev space $H^s(\mathbb S^{d-1})$. However, our spherical function $g_x$ from eqt. (11) has a very special form. In particular, it is symmetric, i.e., $g_x(\xi)=g_x(-\xi)$, such that the integral over the sphere can be replaced by the integral over an arbitrary hemisphere. If we choose this hemisphere properly, then the function $g_x$ is infinitely differentiable on it for all considered kernels, since the non-smoothness in our examples only appears at the equator $\langle \xi, x\rangle=0$.
> We have added a discussion in Appendix K.4 and referenced it in the outlook and limitations in Section 5.
>
> - Regarding Question 1:
> We added an application of computing MMD flows with CIFAR10 (dimension $d=3072$) in the new Appendix K.3, which was used for image generation in the literature. Also in this case, QMC slicing performs better than normal slicing with random directions. RFF-based methods are not applicable here, since the considered negative distance kernel is not positive definite.
>
> - Regarding Question 2:
> The sigmoid kernel is out of the scope of this paper, as it is not a radial one. We included experiments for the thin plate spline kernel in Figure 8, observing a significant improvement of QMC slicing over slicing also here.

---

### Official Review · Reviewer_T9eK · 2024-11-02

**Soundness:** 3
**Presentation:** 2
**Contribution:** 3
**Rating:** 6
**Confidence:** 2

**Summary:**

The paper proposes a slicing based fast calculation of the sum of radial kernels. The authors analyses bounds on slicing error, and ensure a better error rate based on the smoothness result.

**Strengths:**

- The theoretical analysis in the paper seemingly technically sound, though I couldn't follow the proofs.

**Weaknesses:**

- Overall, the paper is difficult to follow those who are not familiar with the topic. Readability can be improved.

- Empirical evaluation is only on purely kernel sum calculations. Showing benefit on actual computation of some learning model would have been convincing.

**Questions:**

- How is the optimization problem (10) is solved in practice? How large is this additional computational cost? Figure 2 contains it? In particular, when P and d is large, it seems expensive.

---

> ### Author Response · Authors · 2024-11-19
> **Response to the Reviewer**
>
> Thank you for your review. Please find below our answers to your comments.
>
> - Regarding readability (weakness 1): We increased the readability at several places, e.g., in making the notation more consistent at some places (for example, we write $g$ instead of $f$ now in Definition 2, in accordance with the notation $g_x(\xi)=f(|(\xi,x)|)$ in (10)), and streamlining the formulation of Theorem 1. Changes are highlighted in blue.
>
> - Regarding applications (weakness 2): We have added a specific use-case of the fast kernel summation for minimizing an MMD functional by simulating a gradient flow in the new Appendix K.3. This was used in several papers for generative modelling and other applications. We note that the exact computation of the kernel sums is computationally intractable here. Also in this case, we observe a significant advantage of the QMC slicing approach.
>
> - Regarding the questions about the minimization of (10), which is eqt (9) in the revised version: We do not consider the computation of the QMC designs as part of the computation time in Fig. 2, since it only depends on the dimension, the number of projections and the Sobolev smoothness. In particular, it is **independent of the data points and the kernel** such that an a-priori estimation seems reasonable. For $P>d$, we minimize (10) numerically by the Adam optimizer. The computation time of this procedure indeed heavily depends on $P$ and $d$ and ranges from a couple of seconds (in $d=3$) up to one hour (for $P=5120$ and $d=784$), but again, this computation is completely independent of the problem itself. For $P\leq d$ (and $s=\frac{d}{2}$), any orthogonal system is a minimizer in (10) as outlined in Appendix H such that the computation can be done in fractions of a second.

---

### Official Review · Reviewer_EyTQ · 2024-11-02

**Soundness:** 2
**Presentation:** 3
**Contribution:** 2
**Rating:** 6
**Confidence:** 4

**Summary:**

This paper proposes a quasi-Monte Carlo (QMC) slicing approach for fast radial kernel summation, which is commonly required in kernel methods across machine learning. By using QMC sequences and spherical quadrature rules, the authors derive error bounds for various kernels, such as the Gauss, Laplace, and Matérn kernels, and show that QMC slicing significantly improves computation time and accuracy. Numerical experiments confirm that QMC slicing outperforms non-QMC slicing and other approximation methods, particularly in lower-dimensional cases, while maintaining favorable error rates.

**Strengths:**

1.The QMC slicing method introduces an approach to fast kernel summation.
2.The methodology is well-documented, with explanations of error bounds and smoothness results.
3.QMC slicing has the potential to improve efficiency in kernel-based methods.

**Weaknesses:**

1.The adaptability of the theoretical error bounds to various types of data and kernels remains unclear.
2.While effective, the QMC slicing method lacks clear differentiation from existing fast summation methods, such as Random Fourier Features (RFF).
3.Experiments are focused on limited, synthetic datasets, raising concerns about the method's generalizability.
4.Some parts of the theory section are overly condensed, especially around error bounds and smoothness assumptions.

**Questions:**

1.How scalable is the QMC slicing method for very high-dimensional data (e.g., 100+ dimensions)?
2.Does the QMC slicing approach generalize to non-standard kernels often used in machine learning?
3.How do theoretical error bounds hold up on real-world datasets with noise and variability?
4.Can the authors compare the computational efficiency of QMC slicing to Random Fourier Features and other fast summation methods?
5.What are the practical trade-offs in terms of error convergence rate with QMC slicing?
6.Are there specific use cases or domains where this approach is expected to outperform current alternatives?

---

> ### Author Response · Authors · 2024-11-19
> **Response to the Reviewer**
>
> Many thanks for your review! We answer to each of your comments separately.
>
> *Adaptability of the theoretical error bounds* (addresses weakness 1): Our error bounds apply to any kind of data in Euclidean space and to any positive definite radial kernel, as well as the Riesz kernel and the thin plate spline. We now mention this in Section 1, Contributions.
>
> *Comparison to RFF* (addresses weakness 2 and question 4): We numerically compare to RFF and related methods (like ORF and QMC-RFF) in our paper (Fig. 1, 2, 5, 6, 7, 8, 9, 10) and demonstrate in all cases a significant advantage of QMC slicing. From a theoretic side, we have outlined in detail (Appendix G), how the slicing method and RFF are related. Some fundamental differences include
>
> - RFFs rely on Bochner's theorem and are consequently only applicable to positive definite kernels. We presented several examples in the paper, where this assumption is violated. In particular, they include the negative distance kernel (Fig. 4, 7, 8, 9, 10), which is widely used in the energy distance (Szeleky 2002). We additionally added the thin plate spline kernel as an example in Figure 8 and added an application to MMD flows in Appendix K.3. In both cases, RFF are not applicable, since the kernel is not positive definite.
>
> - RFF integrate over the measure from Bochner's theorem, while slicing integrates over the unit sphere. In particular, we can exploit QMC designs and quadrature rules on the sphere for QMC slicing, which is a well studied problem from numerical analysis. In contrast, methods combining QMC with RFF always rely on a transformation of the measure from Bochner's theorem to the uniform distribution on the unit cube, which is only possible in very restrictive examples. Due to this difference, our QMC slicing significantly outperforms RFF and related methods like orthogonal Fourier features and QMC RFF (comparisons in Fig. 1, 2, 5, 6, 7, 8, 9, 10).
>
> - We added a theoretical analysis of the complexity of QMC-Fourier-Slicing and RFF for the Gauss kernel in the new Proposition 5 in Appendix J.
>
> *Readability* (addresses weakness 4): We streamlined the formulation of Theorem 1 and made several minor improvements to increase the readability. Taking into account the page limit, it seems unfeasible to add longer explanations to the main text without removing other content.
>
> *Applications* (addresses weakness 3 and question 3): We added an application of computing an MMD flow in the new Appendix K.3, where we show a clear advantage of QMC slicing. Additionally, we would like to highlight that, typically, sensitivity with respect to noise is a property of the specific application and not of how the kernel sums are computed.
>
> *Higher dimensions* (addresses questions 1 and 6): We added an application to MMD flows which acts on the CIFAR10 dataset ($d=3072$) in Appendix K.3. Here, RFF is not applicable because the negative distance kernel is not positive definite, and a direct computation takes considerably more time. Additionally, note that our paper already contained examples in 100+ dimensions: Fig. 9 considers (Fashion)MNIST with $d=784$.
>
> *Other kernels* (addresses question 2): In Fig. 8, we have added the thin plate spline kernel. Note that it is not positive definite such that RFF-based methods are not applicable. Also in this example, we observe a clear advantage of QMC slicing.

---

> > ### Comment · Reviewer_EyTQ · 2024-11-27
> >
> > The authors have answer most of my questions with clear clarifications, addressing my concerns of technical issues. I will improve my scores.

---

### Official Review · Reviewer_WPrq · 2024-11-08

**Soundness:** 3
**Presentation:** 3
**Contribution:** 3
**Rating:** 6
**Confidence:** 4

**Summary:**

In order to reduce the compute time of kernel summations in large dimension, this paper studies 1D slicing when computing kernels on R^d of the form (x,x’) -> F(\| x – x’ \|). The paper is directly built on a work of Hertrich (2024), that introduced the framework of the generalized Riemann-Liouville fractional integral transform to characterize the relationship between the sliced approximation and its target. In a first contribution, a novel error bound is proven with a characterization of the variance of the estimator when Slices are uniformly distributed. In a second contribution, the authors discuss quasi-Monte-Carlo designs (spherical, Sobol) to choose the slices and improve the error rates. The paper is completed by numerical simulation on summations of kernels on artificial data and on MNIST data. While QMC-slicing is interesting in terms of approximation error (convergence rate) their design is expensive and prohibitive for d>= 3.

**Strengths:**

Although slicing has been thoroughly explored in reducing the complexity in time of optimal transport, slicing the computation of kernels is a novel topic, very rencelty introduced.
- The novel error bound differs from Hertrich (2024, SIAM Journal on Mathematics of Data Science and arxiv) with a characterization of the vairance of the slicing estimator, which confirms the rate in O(1/sqrt(P)).
- The most interesting and promising part from my point of view is related to the exploration of quadrature rules to construct the sequence of slices and not sampling it. The proposed scheme (approximation of a spherical design) comes with a bound on the approximation error.

**Weaknesses:**

- About the motivation:The number of data is usually considered as the most emblematic issue with kernels in Machine Learning and existing approximation schemes aim at reducing the compute time involved by operations with the Gram matrix as well as the complexity in memory. The authors motivate their work by the case when the dimension of data is large. Contrary to optimal transport problems that are defined in a variational way,  computing sums of kernels at the era of distributed computing is not a crucial obstacle.

- Limited contribution given previous works:
The most important weakness of the paper is its incrementality with respect to the work of Hertrich (2024) well cited in the paper.
The revisited error bound in the case of the Monte-Carlo estimate does not seem sufficient to make a difference so the new component of the work deals with the proposition of the quasi-Monte-Carlo design for the slicing method.
- No applicabe scale for the spherical deisgn: Unfortunately, the spherical design studied in depth here cannot be reasonably computed for  realistic values of d.
So finally either Monte-Carlo estimation or classic Sobol sequences have to be used in practise (dataset MNIST) which again limits dratsically the novelty of the paper.

In terms of experiments, I would not consider the dimension of MNIST data as a computational obstacle and would definitevely expect higher dimensional data for a conference like ICLR.

Moreover it would have been interesting to apply this new approximated way to compute sums of kernels in a statistical kernel-based test or in an online learning algorithm of a kernel machine where this approximation can make sense. Studying then the convergence of the obtained estimator would have been entirely new.

**Questions:**

1° Can you comment on the interest of slicing in Optimal Transport versus in kernel summation ?
2° It would be interesting to describe the differences point by point of this work and those of Hertrich (2024).
3° Could you clearly describe the analytical complexity in time of each QMC scheme ?
4° Can you find and experiment an example in Machine learning where the sliced kernels can make a difference ?
5° is it possible to run the same tests on larger dimensional data ?

After rebuttal, I've increased my score (most of the concerns have been addressed except those concerning the experimental results.

---

> ### Author Response · Authors · 2024-11-19
> **Response to the Reviewer (1/2)**
>
> We would like to thank the reviewer for the evaluation of our paper. Please find below our detailed answers to the raised comments. Before that, we want to clarify one important misunderstanding regarding the used QMC sequences, which appeared in the summary of the review and in comment 3 of the weaknesses part.
>
> - **The proposed distance QMC designs are applicable for any dimension $d$**, where the minimization problem (9) is tractable, which includes all examples that we considered. We now highlight the use of the distance QMC designs in the "Contributions" in Section 1. They are obtained by minimizing eqt (9) and have a guaranteed convergence rate of $\mathcal O(P^{s/{d-1}})$ for $s\in(\frac{d-1}{2},\frac{d+1}{2})$. However, while the advantage of this convergence rate degrades for large $d$, we **numerically observe much better convergence rates**. This gap between the theoretical guarantees and the numerical observations is discussed in the limitations and, following the comments of the other reviewers, we have added possible reasons and perspectives to overcome it in Appendix K.4.
>
> - Whether Sobol sequences projected to the sphere are QMC designs in the sense of Definition 2 is not clear at all. They were initially built as low discrepancy sequences on $[0,1]^d$. Whether they remain QMC designs after the transformation onto the sphere is unclear (we clarified that in Section 4.1). However, in all our numerical experiments, the Sobol sequence performs better than iid uniformly distributed slices. At the same time, the **Sobol sequence always have a larger error than distance QMC designs**. Therefore, the distance QMC designs are used in Section 4.3.
>
> - In the special case that we can compute spherical designs (which is indeed a hard problem for $d>4$, for $d=4$, they can be computed numerically, see Womersley, 2018), we even get an exponential error rate with a smooth kernel such as the Gaussian.
>
> ## Regarding the motivation, application and high dimensions
>
> (addresses comments 1 and 4 of the weaknesses part and questions 4 and 5)
>
> We disagree with the reviewer that the complexity reduction of kernel summations in the "era of distributed computing is not a crucial obstacle". We added an example of computing an MMD gradient flow, which was used for generative modelling by several papers. In particular, this includes a sequential evaluation of many large kernel sums (similar to an online-learning setting). While for (QMC-)slicing the computation of one step takes **between 0.2 and 0.3 seconds**, the exact evaluation using the high-performance library PyKeOps takes **one hour**. Considering that we run this gradient flow for 50000 steps, this clearly demonstrates the massive need of fast kernel summation methods. Our evaluation also shows the significant advantage of using a QMC sequence instead of the standard slicing approach.
>
> Regarding high dimensions: We do not explicitly motivate our method by high data dimensions $d$, but by the case where the number of data points ($M$ and $N$) is large. We clearly write in the introduction that we mainly target problems in dimension up to 100 or 200, and we strongly believe that enough machine learning problems fall into this dimension range to consider the problem as interesting. On the other side, we agree with the reviewer that high dimensions appear in several applications and therefore consider also some higher-dimensional examples. The added application on MMD gradient flows now works in more than 3000 dimensions. Finally, we stress that common frameworks for the brute-force computation of large kernel sums have strict limitations on the dimensions. For instance, the widely used package KeOps (Charlier et al, 2021, we compared with its Python interface PyKeOps) states in its documentation that the performance significantly drops for dimensions $d>100$ due to register spilling, see the paragraph limitations in https://www.kernel-operations.io/keops/introduction/why_using_keops.html (we also observe this in our experiments with CIFAR10).

---

> ### Author Response · Authors · 2024-11-19
> **Response to the Reviewer (2/2)**
>
> ## Differences to (Hertrich, 2024)
>
> (addresses comment 2 of the weaknesses part and question 2)
>
> We want to highlight two significant contributions of our paper above the mentioned paper.
>
> First, our error bounds for the (non-QMC) slicing are significantly stronger than in (Hertrich, 2024). We stress here that the issue was not about confirming the $\mathcal{O}(P^{-1})$ rate for the MSE (i.e. $\mathcal{O}(1/\sqrt{P})$ for the RMSE/absolute error) of standard slicing, but to clarify how the constant depends on the dimension of the data space.
>
> - For positive definite kernels and Riesz kernels, our error bounds are dimension-independent. In contrast, (Hertrich, 2024) only proves a dimension-independent absolute error bound for the Gauss kernel and conjectures that this should be true for the Laplace and Matern kernel as well. For the Riesz kernel, the error bound in (Hertrich, 2024) depends on the dimension by $\mathcal O(\sqrt{d})$.
>
> - For the thin plate spline kernel, there did not exist an error bound at all before.
>
> - For kernels with analytic basis functions, for the Riesz kernel and for the thin plate spline, we exactly compute the variance, which implies that there do not exist any tighter estimations of the error. For the Gauss kernel, this yields an improvement of the absolute error bound given in (Hertrich, 2024), as we observe that this bound actually decays proportional to $\|x\|^{-2}$ for $x\to\infty$.
>
> Second, we exploit QMC sequences to improve the error rate of $\mathcal O(1/\sqrt{P})$. We prove smoothness results to ensure the applicability and demonstrate by several numerical examples that QMC slicing, most notably the proposed distance QMC designs, significantly outperforms non-QMC slicing and RFF-based methods.
>
> We added the point-by-point comparison of our error bounds to the ones in (Hertrich, 2024) in Appendix D.1.
>
>
> ## Slicing in OT vs kernel summation
>
> (addresses question 1)
>
> In the optimal transport literature, the sliced Wasserstein distance appeared since it is faster to compute than the standard Wasserstein distance. However, it is important to note that the sliced Wasserstein distance is not the same as the Wasserstein distance and its properties fundamentally differ (different sampling complexity, not a geodesic space, the equivalence estimate between Wasserstein and sliced Wasserstein depends exponentially on the dimension). In contrast, in slicing of kernels, the one-dimensional kernels are constructed such that the sliced kernel summation is an unbiased and consistent estimator of the non-sliced one, and the oscillations are independent of the dimension in many cases, as we proved.
>
> We added the sliced Wasserstein distance to the related work section in the introduction.
>
> ## Complexity of QMC Slicing
>
> (addresses question 3)
>
> We describe in detail the computational complexity of the RFF and Fourier slicing in the new Appendix J, and comment on this in Section 3.2. Since both methods have different parameters, we also compared the complexity to achieve a certain accuracy level $\varepsilon$ for the case of the Gauss kernel in Proposition 5.
> There we see that the asymptotic complexity of the QMC Fourier slicing is indeed lower.

---

> > ### Comment · Reviewer_WPrq · 2024-11-25
> > **Comments on  the answers**
> >
> > Thank you for addressing most of my concerns.
> >
> > I acknowledge the motivation was correctly introduced in the paper and I fully understand that the problem is of importance for using  instance the kernel summation in a loss (good point).
> > I also think the point-by-point comparison between the authors contribution on errors made by Non-QMC and then QMC methods with Hertrich's work  (MC) is useful and appreciate the changes.
> > The short discussion about slicing in kernel summation and slicing in OT is useful as well.
> > I am happy to update my score consequently: I think this work together with Hertrich's work will serve as a strong basis for even further development of slicing methods for kernels.

---

### Author Response · Authors · 2024-11-19
**Rebuttal**

We would like to thank all reviewers for the thorough evaluation of our paper. We carefully revised it and give a detailed answer for each reviewer separately. Changes of the paper are indicated in blue. In particular:

- We added an application of (QMC-)Slicing to the computation of MMD gradient flows in Appendix K.3 employing the negative distance kernel. The example operates on the CIFAR10 dataset in $d=3072$ dimensions. We clearly see the advantage of QMC slicing over non-QMC slicing. Additionally, we note that the exact evaluation of the kernel sums is intractable in these cases, and Random Fourier Features are not applicable due to the fact that the considered kernel is not positive definite.

- As requested by some of the reviewers, we added an additional kernel (thin plate spline kernel) to Figure 8.

- We added a summary of the computational complexity for achieving a certain error with QMC slicing and compare it with the error of Random Fourier features in Proposition 5.

In addition, we incorporated many smaller corrections and improvements according to the detailed comments of the reviewers.

---

### Meta-Review · Area_Chair_ZUpQ · 2024-12-20

**Metareview:**

The paper investigates the problem of fast computation of large kernel sums via the method of slicing, which relies on random projections to one-dimensional subspaces and fast Fourier summation. In particular, the authors prove bounds for the slicing error and propose a quasi-Monte Carlo (QMC) approach for selecting the projections based on spherical quadrature rules.

Reviewers generally appreciate the proposed method, its theoretical analysis, and potential in improving the efficiency of kernel-based methods.

**Additional Comments On Reviewer Discussion:**

- Reviewer WPrq pointed out that a major weakness of the paper is its incremental novelty with respect to the previous work (Hertrich 2024), which was frequently cited. The authors responded by pointing out the improvement in their error bounds, which are now dimension-independent.

- Reviewer  WPrq, EyTQ expressed concerns about the scalability to high dimension. In their rebuttal, the authors responded by adding an application of computing MMD flows with CIFAR10 (dimension 3072)

-  Reviewer  WPrq: the spherical design is not tractable for dimension $d > 3$, which drastically reduces the novelty of the paper. The authors also acknowledged as their limitations that the advantage of QMC slicing become smaller in higher dimensions.

Despite these weaknesses, I believe the reported results are of sufficient interest to recommend acceptance.

---

### Decision · Program_Chairs · 2025-01-22

Accept (Poster)